# Risk of Fat Mass- and Obesity-Associated Gene-Dependent Obesogenic Programming by Formula Feeding Compared to Breastfeeding

**DOI:** 10.3390/nu16152451

**Published:** 2024-07-28

**Authors:** Bodo C. Melnik, Ralf Weiskirchen, Wolfgang Stremmel, Swen Malte John, Gerd Schmitz

**Affiliations:** 1Department of Dermatology, Environmental Medicine and Health Theory, University of Osnabrück, D-49076 Osnabrück, Germany; 2Institute of Molecular Pathobiochemistry, Experimental Gene Therapy and Clinical Chemistry (IFMPEGKC), RWTH University Hospital Aachen, D-52074 Aachen, Germany; rweiskirchen@ukaachen.de; 3Praxis for Internal Medicine, Beethovenstrasse 2, D-76530 Baden-Baden, Germany; wolfgangstremmel@aol.com; 4Institute for Interdisciplinary Dermatological Prevention and Rehabilitation (iDerm), University of Osnabrück, D-49076 Osnabrück, Germany; sjohn@uos.de; 5Institute for Clinical Chemistry and Laboratory Medicine, University Hospital of Regensburg, D-93053 Regensburg, Germany; gerd.schmitz@ukr.de

**Keywords:** adipogenesis, breastfeeding, DNA methylation, epigenetic stem cell regulation, fat mass- and obesity-associated gene (FTO), formula feeding, m6A RNA methylation, milk exosomal miRNAs, wingless signaling, obesity

## Abstract

It is the purpose of this review to compare differences in postnatal epigenetic programming at the level of DNA and RNA methylation and later obesity risk between infants receiving artificial formula feeding (FF) in contrast to natural breastfeeding (BF). FF bears the risk of aberrant epigenetic programming at the level of DNA methylation and enhances the expression of the RNA demethylase fat mass- and obesity-associated gene (*FTO*), pointing to further deviations in the RNA methylome. Based on a literature search through Web of Science, Google Scholar, and PubMed databases concerning the dietary and epigenetic factors influencing *FTO* gene and FTO protein expression and FTO activity, FTO’s impact on postnatal adipogenic programming was investigated. Accumulated translational evidence underscores that total protein intake as well as tryptophan, kynurenine, branched-chain amino acids, milk exosomal miRNAs, NADP, and NADPH are crucial regulators modifying *FTO* gene expression and FTO activity. Increased FTO-mTORC1-S6K1 signaling may epigenetically suppress the WNT/β-catenin pathway, enhancing adipocyte precursor cell proliferation and adipogenesis. Formula-induced FTO-dependent alterations of the N6-methyladenosine (m6A) RNA methylome may represent novel unfavorable molecular events in the postnatal development of adipogenesis and obesity, necessitating further investigations. BF provides physiological epigenetic DNA and RNA regulation, a compelling reason to rely on BF.

## 1. Introduction

According to the “Developmental Origins of Health and Disease (DOHaD)” hypothesis, the postnatal period plays a key role in developmental programming, conditioning obesity, diabetes, and metabolic syndrome later in life [1]. The metabolic environments during the prenatal, postnatal, and early childhood periods are modified by maternal nutrition, postnatal feeding, and early childhood adiposity, all having an impact on health outcomes in adult life [1]. In contrast to formula feeding (FF), breastfeeding (BF) involves the physiological nutrition and mammary gland-mediated epigenetic programming of the infant [2,3,4,5,6,7]. It offers protective metabolic programming against chronic non-communicable diseases such as childhood and adult obesity [8,9,10,11,12,13,14,15,16,17,18,19,20,21] as well as type 2 and type 1 diabetes mellitus [9,22,23,24,25]. Nutrition is one of the most influential factors directly impacting DNA methylation pathways [26]. A large percentage of tissue-specific methylation patterns are generated postnatally [27]. The postnatal period, during which human infants are physiologically exposed to maternal breastmilk, appears to be a critical window for determining long-term metabolic health [28]. Nutrition in early life modifies the epigenome [29], leading to different phenotypes and altering disease susceptibilities [30]. Breastmilk is evolution’s optimized adaptation for this critical developmental period of human life, correctly adjusting the epigenetic axes through milk’s bioactive compounds, including milk exosomal microRNAs (miRNAs) [1,2,3,4,5,6,7].

## 2. Materials and Methods

To investigate the potential impact of fat mass- and obesity-associated gene (*FTO*) overexpression through formula feeding and to identify metabolic and dietary factors that modify *FTO* expression and their impact on RNA methylation patterns involved in lipogenesis and adipogenesis, Web of Science, Google Scholar, and PubMed data bases were selected to choose research articles all published in the English language. The scientific quality and validity of the papers were controlled by all contributing authors. The following key words were used for the search: “fat mass- and obesity-associated protein”, “FTO”, “infant formula”, “formula feeding”, “human milk”, “breastmilk”, “breastfeeding”, “microRNA”, “miRNA”, “miR”, “epigenetic regulation”, “DNA methylation”, “RNA methylation”, “N6-adenosine methylation”, “m6A”, “m6A RNA demethylation”, “lipid synthesis”, “lipid storage”, “lipogenesis”, “adipocyte”, “adipogenesis”, “obesity”, “obesity risk”, “body mass index”, “BMI”, “fat”, “fat mass”, “lean mass”, “white adipose tissue”, “WAT”, “brown adipose tissue”, “BAT”, “beige adipose tissue”, “thermogenesis”, “energy expenditure”, “food intake”, “hypothalamic control”, “satiety”, “hyperphagia”, “circadian clock”, “circadian rhythm”. Papers presenting redundant information from the same research facility were excluded from this review. A total of 343 papers were assessed.

## 3. Results

### 3.1. Breastfeeding in Comparison to Infant Formula Feeding Modifies DNA Methylation

The programmed postnatal changes in methylation that occur in various tissues of the body are crucial for proper postnatal development and tissue maturation [27]. It is important to note that whether methylation inhibits or activates transcription depends on the gene segment being analyzed. Hypermethylation in gene bodies is linked to gene expression [31], while hypermethylation at transcription start sites or enhancers always results in gene silencing [32].

Accumulating evidence supports the concept that BF and the duration of BF result in substantial variations in DNA CpG island methylation of gene promoters, enhancers, and gene bodies compared to artificial FF. CpG islands are DNA methylation regions in promoters that silence corresponding gene expression.

The duration of BF was negatively associated with methylation of the leptin gene (*LEP*) in the whole blood of children at 17 months of age [33]. Sherwood et al. [34] confirmed that BF duration correlates with *LEP* methylation at the age of 10 years and with body mass index (BMI) trajectory. According to a recent systematic review [35], BF negatively correlates with promoter methylation of *LEP*, *CDKN2A,* and *SLC2A4* genes and is positively associated with *NYP* promoter methylation. BF may also modify global methylation patterns and epigenetic effects of some genetic variants. In line with this, BF duration was associated with epigenetic variations in *RXRA* and *LEP* at the age of 12 months, as well as infant growth trajectories [36].

An epigenome-wide association study revealed that the duration of BF is associated with epigenetic differences in children [37]. DNA methylation data derived from the ALSPAC child cohort [38] reported that exclusive BF exhibits a higher number of DNA methylation variations during infancy than at other growth periods of childhood. At the genome-wide level, 13 CpG sites in girls (*MIR21*, *SNAPC3*, *ATP6V0A1*, *DHX15*/*PPARGC1A*, *LINC00398*/*ALOX5AP*, *FAM238C*, *NATP*/*NAT2*, *CUX1*, *TRAPPC9*, *OSBPL1A*, *ZNF185*, *FAM84A*, *PDPK1*) and 2 CpG sites in boys (*IL16* and *NREP*) mediate the association between exclusive BF and longitudinal BMI. Taken together, exclusive and prolonged BF in contrast to FF could play a major role in preventing the risks of overweight/obesity in children and adults through DNA methylation mechanisms occurring in postnatal life. These observations clearly confirm milk’s impact on epigenetic regulation [39].

Recently, Trinchese et al. [40] reported significant variations in epigenetic modulations in the brains of rats following the administration of milk from different animal species (cow, human, and donkey). The milk intake of different mammals induced relative differences in promoter methylation levels (cow 40%, human 43%, and donkey 44%) as well as methylation levels of gene bodies (cow 47%, human 60%, and donkey 56%), leading to variations in gene expression. These data point to differences in the capacity and magnitude of promoter and whole gene body methylations between cow milk and human milk intake. Unfortunately, the investigators did not include cow milk-derived artificial formula in their study.

### 3.2. Variations in DNA CpG Methylation Modify FTO and CUX1 Gene Expression

DNA methylation at CpG sites of the fat mass and obesity gene (*FTO*) is important for regulating the level of *FTO* expression and the effects of individual single nucleotide polymorphisms (SNPs) of *FTO* [41,42,43,44]. Bell et al. [45] reported haplotype-specific methylation in the *FTO* type 2 diabetes and obesity susceptibility locus, with measurable methylation differences within the SNP association locus in a genome-wide association study. Czogała et al. [46] found that obese children showed significantly higher expression and methylation levels of the *FTO* gene compared to healthy controls. Furthermore, *FTO* gene expression and *FTO* methylation correlated with changes in circulating levels of adipokines and gastrointestinal peptides in children [44]. The expression level of FTO showed a negative correlation with the expression of apelin and leptin receptors, but a positive association with leptin expression. Additionally, both *FTO* methylation and expression negatively correlated with resistin and visfatin. Whereas cholecystokinin negatively correlated with *FTO* methylation and expression, fibroblast growth factor 21 showed a positive correlation. Moreover, FTO expression exhibited a negative association with cholecystokinin and glucagon-like peptide-1 levels.

SNPs in the first intron of *FTO* are robustly associated with BMI and obesity [45,46,47,48,49,50]. The 3.4 kb region upstream of the human *FTO* contains a transcriptional initiation site for the retinitis pigmentosa GTPase regulator interacting protein 1-like (*RPGRIP1L*) gene. The first intron of *FTO* harbors most obesigenic SNPs and contains a binding site for the transcription factor cut-like homeobox 1 (*CUX1*). CUX1 promotes the expression of *RPGRIP1L* [49,50]. Specific subsets of hypothalamic neurons attenuate lipolysis by regulating *FTO* expression [51]. CUX1 interacts with the binding site of *FTO* intron 1, upregulating the adjacent *RPGRIP1L* and inhibiting leptin receptor transport and leptin signaling [52,53,54].

*RPGRIP1L* is located more than 100 base pairs away in the opposite transcriptional orientation of *FTO*. It is involved in anchoring the complex to promote leptin receptor aggregation, regulating cellular leptin sensitivity and food intake [53]. Increased hypothalamic FTO activity promotes leptin resistance, enhancing caloric intake and reducing lipolysis [55]. CUX1-mediated upregulation of RPGRIP1L by binding to *FTO* intron 1 may reduce leptin sensitivity and lipolysis, promoting the development of obesity. Notably, *CUX1* has been identified as one of the genes exhibiting substantial CpG DNA methylation variations during infancy compared to other growth periods during childhood in girls, linking exclusive BF to longitudinal BMI [38].

### 3.3. The Epigenetic Impact of RNA N6-Methyladenosine Modifications on Adipogenesis

The m6A modifications are the most abundant methylation changes in mRNAs and non-coding RNAs (ncRNA), affecting many aspects of mRNA metabolism, including splicing, translation, stability, and miRNA maturation. They play critical roles in gene regulation, influencing various cellular processes [56,57,58,59]. m6A modifications are inserted by RNA methyltransferases METTL3, METTL14, and WTAP (called writers), are removed by the demethylases FTO and ALKBH5 (erasers), and are recognized by m6A binding proteins, including YT521-B homology YTH domain-containing proteins (readers) [56,57,58,59]. Moreover, m6A modifications have attracted recent attention due to their pivotal importance in the regulation of adipogenic gene expression [60,61,62].

Recent studies have shown that m6A marks are decreased in the adipose tissue of obese subjects. They are involved in the regulation of obesity-associated processes, including adipogenesis, regulation of lipid metabolism, and insulin resistance [50,63,64].

### 3.4. Formula Feeding Compared to Breastfeeding Increases FTO Expression

Accumulated evidence demonstrates that FF compared to BF modifies epigenetic changes at the level of CpG DNA methylation. Furthermore, excessively overexpressed *FTO* in peripheral blood mononuclear cells of FF infants compared to exclusively BF infants at the age of 5 to 6 months has recently been reported [65]. The group of exclusively BF infants exhibited the lowest level of *FTO* gene expression (3.39 ± 1.1) compared to the FF group of infants (89.2 ± 19.3) (*p* < 0.001). A significant increase in *FTO* gene expression compared to exclusive BF was also observed in the group of infants receiving mixed feeding (BF and FF) (59.3 ± 9.3) [65]. In contrast to exclusive BF, the expression of peroxisome proliferator-activated receptor-α (*PPARA*) (85.41 ± 17.8) was significantly reduced in infants receiving mixed feeding (23.6 ± 10.98) and FF (19.4 ± 5.6), respectively. Recent evidence indicates that *FTO* overexpression as well as reduced m6A-RNA methylation suppresses *PPARA* [66,67], a critical transcription factor orchestrating anti-obesigenic and anti-inflammatory metabolic effects [68].

Currently, research on FTO’s role in obesity is divided into two groups. One group hypothesizes that *FTO* introns serve as cis-regulatory sites for nearby genes involved in obesity risk, while the other group suggests that *FTO* introns regulate *FTO* gene expression itself [62]. It is worth mentioning that *FTO* SNPs in intron 1 (rs9939609, rs17817449, rs3751812, rs1421085, rs9930506, and rs7202116) are strongly associated with overweight or obesity [69,70].

Recent evidence supports the fact that the m6A demethylase FTO [71,72] directly modulates obesity through m6A RNA modifications [62]. Remarkably, the most prevalent SNP rs9939609 risk allele is associated with increased *FTO* mRNA levels [73]. Increased expression of FTO contributes to the obesity phenotype, as demonstrated in mice with two additional copies of *FTO* (FTO-4 mice) showing increased adiposity and hyperphagia [74]. Berulava et al. [75] provided evidence that obesity-associated SNPs in intron 1 of the *FTO* gene enhance primary transcript levels of *FTO*. Specifically, *FTO* transcripts containing the A (risk) allele of rs9939609 were more abundant than those with the T allele (mean 1.38; 95% confidence interval (CI) 1.31–1.44) [75].

A longer duration of exclusive BF (at least 5 months) has a significant influence on BMI growth trajectories among children carrying the adverse FTO variant by modulating the age at adiposity peak, age at adiposity rebound, and BMI velocities. Notably, exclusive BF acts antagonistically to the *FTO* rs9939609 risk allele, and by the age of 15 years, the predicted reduction in BMI after 5 months of exclusive BF is 1.14 kg/m^2^ (95% CI: 0.67–1.62; *p* < 0.0001) and 0.56 kg/m^2^ (95% CI: 0.11–1.01; *p* = 0.003) in girls and boys, respectively [76]. Evidence from longitudinal cohort studies shows that exclusive BF versus FF/mixed feeding or longer BF duration is associated with lower BMI trajectories [77]. Such early interventions became more apparent with age and were sustained into early adult life [77]. These findings imply that artificial FF may not meet the appropriate physiological FTO axis naturally controlled by BF. FTO plays a key role in regulating the mRNA and miRNA methylome involved in posttranscriptional gene expression, including alternative splicing. Excellent recent publications provide extensive information on FTO and its impact on m6A-dependent RNA expression [78,79,80,81,82,83,84]. Table 1 summarizes recent evidence for FTO-mediated m6A RNA demethylation involved in the regulation of lipid metabolism, adipogenesis of white (WAT) and brown adipose tissue (BAT), and satiety control.

The accumulated evidence presented in Table 1 leads to the conclusion that over-activated FTO demethylase activity promotes pre-adipocyte and adipocyte differentia-tion of WAT [88,95,96], increases lipid synthesis of WAT [61,85,86,87,88,89,90,91,92,93,94,95,96,97], and promotes ghrelin-induced hyperphagia [73]. Conversely, suppression of FTO activity is related to the development of BAT, involved in thermogenesis and energy expenditure [98,99,100,101,102].

The following chapters analyze pathways that potentially enhance FTO overacti-vation by infant FF and their aberrant effects on gene expression. Table 2 represents a selection of adipogenesis-related genes modified by FTO-mediated m6A RNA demeth-ylation.

### 3.5. High Protein Formula Intake and the Risk of FTO and mTORC1/S6K1 Overactivation

The European Childhood Obesity Trial Study Group enrolled healthy term infants from five European countries in a double-blind randomized trial with a follow-up period of 11 years [124]. Compared to conventional high-protein formula, feeding infants a lower protein formula lowered BMI trajectories up to 11 years and achieved similar BMI values at adiposity rebound as observed in BF infants. BMI trajectories were increased in the higher protein group compared to the lower protein group, with a significantly different BMI at adiposity rebound (0.24 kg/m^2^, 0.01–0.47, *p* = 0.040) and an increased risk for overweight at 11 years (adjusted Odds Ratio (OR) 1.70; 1.06–2.73; *p* = 0.027) [124]. This study confirms earlier observations linking high-protein formula intake to later weight gain and obesity risk [125,126,127,128,129].

Accumulated evidence links protein and amino acid (AA) availability to the activation of FTO. Doaei et al. [130] explored the interactions between protein intake and *FTO* gene expression in obese and overweight male adolescents after 18 weeks of intervention. They found that a higher protein intake significantly upregulated the *FTO* gene [130]. FTO functions as an AA sensor, connecting AA levels to the activity of mammalian target of rapamycin complex 1 (mTORC1) signaling, thereby playing a crucial role in regulating growth and translation [131,132,133]. FTO resides in the nucleus and cytoplasm, with a mobile fraction shuttling between both cell compartments, possibly mediated by exportin 2 [134]. In vivo experiments have shown that FTO is involved in linking AA availability to the mTORC1 signaling pathway. Cells lacking FTO exhibit reduced activation of the mTORC1 pathway, slower mRNA translation, and increased autophagy [131,132]. The proposed model of Gulati et al. [132,133] predicts that FTO operates upstream of amino-acyl tRNA synthetases (AARS) and leucyl-tRNA synthase (LARS) to connect AA levels with mTORC1 signaling [135,136]. LARS plays a crucial role in AA-induced mTORC1 activation by sensing intracellular leucine levels and initiating events that lead to mTORC1 activation. LARS directly interacts with Rag GTPase, the mediator of AA signaling to mTORC1, in an AA-dependent manner, acting as a GTPase-activating protein (GAP) for Rag GTPase to activate mTORC1 [133,134]. There is current uncertainty concerning the potential impact of FTO on *LARS1* RNA m6A demethylation, which could potentially enhance *LARS1* expression under conditions of activated FTO. Intriguingly, FTO enhances the expression of activating transcription factor 4 (ATF4) [107], a key transcriptional regulator of AA uptake and metabolism and downstream activator of *LARS* expression [137]. Therefore, FTO activation enhances ATF4-LARS-mTORC1 signaling, establishing a significant biological pathway linking FTO and mTORC1 signaling (Figure 1).

A recent study on weaning piglets identified a regulatory influence of dietary BCAAs on lipid metabolism, partially through changes of m6A RNA methylation [138]. Intriguingly, activated FTO decreases gene expression of TSC complex subunit 1 (*TSC1*), while knockdown of *FTO* increases the mRNA level of *TSC1* [139]. The TSC1/TSC2 complex is a critical molecular checkpoint, inhibiting mTORC1 activation [140]. Activation of mTORC1 increases mRNA and protein expression of peroxisome proliferator-activated receptor-γ (PPARγ) and sterol-regulatory element binding transcription factors (SREBFs) [141], master transcriptional regulators of adipocyte differentiation and lipogenesis [142,143] (Table 2). *FTO* overexpression with recombinant adenovirus encoding the human *FTO* genome in human myotubes resulted in significant overexpression of PPARγ [144].

As shown in hepatocytes, FTO-mediated m6A demethylation increases the expression of ATF4 [105], which cooperates with CCCTC-binding factor (CTCF) to control adipogenesis and adipose development by orchestrating the transcription of adipogenic genes such as *CEBPD* and *PPARG*, co-regulating their transactivation [145]. Notably, CTCF binds to an enhancer of the *FTO* gene, increasing *FTO* expression [146]. ATF4-deficient mice exhibit increased energy expenditure, enhanced lipolysis, upregulation of uncoupling protein 2 (UCP-2) and β-oxidation genes, and decreased expression of lipogenic genes in WAT, suggesting enhanced utilization but attenuated synthesis of fatty acids. Expression of UCP-1, 2, and 3 was also enhanced in BAT, indicating increased thermogenesis. The effect of *ATF4* deletion on UCP expression in BAT leads to the conclusion that increased thermogenesis may underlie raised energy expenditure [147]. In accordance, adult-onset agouti-related peptide neuron-specific *Atf4* knockout (AgRP-ATF4 KO) mice are lean, exhibiting improved insulin and leptin sensitivity and reduced hepatic lipid accumulation. Furthermore, AgRP-ATF4 KO mice exhibit reduced food intake and increased energy expenditure, resulting from increased thermogenesis in BAT. Moreover, AgRP-ATF4 KO mice are resistant to high-fat diet (HFD)-induced obesity, insulin resistance, and liver steatosis and under cold stress maintain a higher body temperature. In hypothalamic GT1-7 cells, the expression of *FOXO1* is directly regulated by ATF4 via binding to the cAMP-responsive element site on its promoter [148]. Drosophila melanogaster mutant flies with insertions at the *ATF4* locus exhibit reduced fat content, increased sensitivity to starvation, and lower circulating carbohydrate levels [149].

Notably, *Atf4* null mice exhibit reduced expression of genes regulating intracellular AA concentrations and lower intracellular AA levels, which under normal conditions function as crucial stimuli in activating mTORC1 [150,151,152]. Of note, *Atf4* null mice exhibit reduced S6K activity in both liver and adipose tissues [149]. Remarkably, mTORC1 controls ATF translation and mRNA stability [153]. Of importance, mTORC1 upregulates the transcription of AA transporters, metabolic enzymes, and aminoacyl-tRNA synthetases, a regulatory program mediated through posttranscriptional control of *ATF4* [153]. Thus, FTO-mediated upregulation of ATF4 may have a direct impact on adipogenesis and cellular AA uptake, a critical input signal for mTORC1/S6K1 activation [150,151,152].

mTORC1 activation is enhanced not only by the AA pathway but also by insulin/IGF-1/PI3K-AKT signaling [154]. Experimental evidence supports the idea that reduced expression of the TSC1/TSC2 complex enhances insulin/AKT/mTORC1/PPARγ-stimulated adipocyte differentiation [155]. Another crucial transcription factor for adipocyte differentiation is CCAAT/enhancer-binding protein β (C/EBPβ) [156]. The translational adjustment of C/EBPβ-isoform expression is a crucial process, directing adaptive responses to changing mTORC1 activity [157]. It is worth noting that the translation of C/EBPβ mRNA into the C/EBPβ-LIP isoform is attenuated in response to mTORC1 inhibition [157]. Interestingly, the FTO protein is a transcriptional coactivator that enhances the transactivation potential of C/EBPs from unmethylated as well as methylation-inhibited gene promoters [158]. The coactivator role of FTO in modulating the transcriptional regulation of adipogenesis by C/EBPs corresponds to the progressive loss of adipose tissue in FTO-deficient mice [159]. FTO deficiency is associated with a significant reduction in body weight and fat mass, particularly of WAT but not BAT. Additionally, FTO deficiency promotes white adipocyte transformation into brown/beige adipocytes [159].

Taken together, FTO is critically involved in mTORC1-PPARγ-C/EBPβ signaling, which plays a critical role in early adipocyte precursor commitment, preadipocyte differentiation, adipocyte maturation, triacylglycerol synthesis, and adipocyte mobilization [160,161,162]. Of importance, the phosphorylated kinase S6K1, the downstream target of activated mTORC1, promotes early adipocyte differentiation and is involved in the commitment of embryonic stem cells to early adipocyte progenitors [163]. Remarkably, a lack of S6K1 in HFD-challenged mice impairs the generation of de novo adipocytes, consistent with a reduction in early adipocyte progenitors [163]. In fact, early adipogenic programming by high-protein formula intake in contrast to low-protein formula intake has been linked to increased AA-mediated mTORC1/S6K1 signaling [164,165]. HEK293 cells transiently transfected with vectors expressing GFP-WT FTO resulted in a dose-dependent increase in S6K1 phosphorylation, whereas AA deprivation reduces FTO levels and S6K1 phosphorylation in empty vector-transfected HEK293 cells [133].

Recently, Yi et al. [166,167] demonstrated that activated S6K1 moves into the nucleus and phosphorylates histone protein H2BS36, subsequently allowing the recruitment of the histone-lysine *N*-methyltransferase enhancer of zeste homolog 2 (EZH2), resulting in H3K27 trimethylation (H3K27me3). This epigenetic switch suppresses the expression of *WNT6*, *WNT10A,* and *WNT10B* genes, leading to adipogenic commitment through the final expression of adipogenic transcription factors PPARγ and C/EBPα. WAT from S6K1-deficient mice consistently exhibit marked reduction in H2BS36 phosphorylation (H2BS36p) and H3K27me3, leading to enhanced expression of *WNT* genes. In addition, expression levels of H2BS36p and H3K27me3 are highly elevated in WAT from mice fed an HFD or from obese humans [166,167]. WNT signaling mainly plays a key role in suppressing adipogenesis during several stages of differentiation. Mesenchymal stem cell (MSC)-derived adipogenesis involves two steps: firstly, MSC differentiation into proliferative preadipocytes and secondly, their terminal differentiation into mature adipocytes. The initiation of both steps requires the suppression of WNT/β-catenin signaling. WNT/β-catenin signaling plays a well-known role in stem cell regulation. Attenuated WNT signaling increases the cell mass of MSCs and preadipocytes [168]. In fact, it has been demonstrated that blocking the WNT signaling downstream of WNT6, WNT10A, and WNT10B promotes differentiation into mature adipocytes, whereas increased expression of WNT6, WNT10A, and WNT10B prevents adipogenesis [169,170,171].

In addition, S6K1 controls adiponectin expression by inducing a transcriptional switch from BMAL1 to EZH2 [171]. Active S6K1 via induction of the suppressive histone code cascade, H2BS36p-EZH2-H3K27me3, also suppresses adiponectin expression. Active S6K1 phosphorylates BMAL1, an important transcription factor regulating the circadian clock system, and dissociates BMAL1 from the *ADIPOQ* promoter region, resulting in EZH2 recruitment and subsequent H3K27me3 modification of the *ADIPOQ* promoter [172]. Recent evidence points to an important role of adiponectin in stimulating Sca1^+^CD34^−^-adipocyte precursor cells associated with hyperplastic expansion and browning of white adipose tissue [173]. Conversely, inhibition of S6K1 hampers fat mass expansion and mitigates HFD-induced hepatosteatosis [174].

Postnatal overactivation of FTO-mTORC1-S6K1 signaling by FF and deficiency of milk miRNA-148a-3p and miRNA-22-3p via deficient DNMT1 and DNMT3A suppression may converge in decreased *WNT* gene expression. On the other hand, both miRNAs directly target *WNT10B* mRNA in a counter-regulatory mode [175,176]

A potential net decrease in WNT signaling due to FF with excessive protein intake and the absence of FTO- and ZFP217-targeting milk miRNAs may finally enhance the total number of adipocyte precursor cells, a critical disturbance during postnatal adipocyte development, priming early steps of aberrant adipogenesis, which may increase the risk of obesity later in life (Figure 2).

A complex networking of epigenetic mechanisms controls WNT and MYC signaling, promoting adipocyte precursor cell proliferation and mass expansion, a fundamental step for postnatal adipogenesis and obesogenic programming of the newborn infant.

In contrast to BF, overactivation of FTO and mTORC1 signaling due to high-protein FF during the postnatal period may lead to a lifelong increase in adipocyte numbers [163,164,165]. This can result in excessive fat accumulation during puberty, especially when combined with exposure to an HFD, explaining the obesity rebound typically seen at age 11 in response to excessive protein consumption during early infancy [124]. An analysis from 14 studies including 16,094 boys and girls aged 1–18 years revealed that the gain-of-function *FTO* rs9939609 variant is linked to a higher BMI and increase of total energy intake. However, lower dietary protein intake mitigated the connection between the FTO genotype and adiposity during childhood and adolescence [177].

### 3.6. The Potential Adipogenic Impact of Exosomal Milk miRNA Deficiency in Infant Formula

Exclusive BF, in comparison to FF, not only limits protein intake and milk protein-dependent *FTO* expression but also provides a significant amount of milk exosomes containing gene-regulatory miRNAs [178,179,180,181,182,183,184,185]. Human milk is the most enriched source of miRNAs among all human body fluids [186], transferring an estimated amount of 176 trillion exosomes daily to the infant though the consumption of 800 mL of breastmilk [187]. Unlike human milk, formulas contain neither exosomes nor milk miRNAs, or they are only present in very low concentrations [188,189]. Abbas et al. [190] recently presented translational evidence supporting the regulation of adipogenesis by exosomal milk miRNAs.

Higher *FTO* expression compared to healthy controls has been linked to significantly increased methylation values for the *FTO* gene [46]. Research has shown that DNA methyltransferase 1 (*DNMT1*) is a direct target of milk exosomal miRNA-148a-3p [191,192], which is a crucial component of milk exosomes [193,194]. In comparison to mature human milk, levels of miRNA-148a-3p in infant formula were less than 1/500th [195]. Remarkably, DNA methylation appears to exert a biphasic regulatory role in adipogenesis, promoting differentiation at the early stage while inhibiting lipogenesis at the late stage of 3T3-L1 preadipocyte differentiation [196]. DNMT1 or DNMT3A overexpression at the beginning of the differentiation promotes 3T3-L1 adipogenesis and enhanced the expression of adipogenic markers such as PPARγ2, C/EBPα, and adipocyte protein 2 (aP2). Conversely, inhibiting DNA methylation by 5-aza-dC at the early differentiation period inhibited adipogenesis, evident by decreased expression of adipogenic markers and adipocyte phenotypic genes and by overexpression of *WNT10A* [196]. Milk exosomal miRNA-148a-3p targets *DNMT1* mRNA, whereas milk exosomal miRNA-22-3p/miRNA-30b-5p/miRNA-200bc-3p/miRNA-29b-3p collectively target *DNMT3A* mRNA [197], thus limiting adipocyte lineage commitment and reducing the numbers of adipocyte precursors at a critical early window of adipogenesis.

miRNA-21-5p, a dominant miRNA found in human milk [180], indirectly downregulates the expression of DNMT1 [198]. The absence of milk exosomal miRNA-148a-3p/miRNA-21-5p in formula, which inhibit DNMT1, may lead to increased methylation of the *FTO* gene and higher FTO expression. Many experimental studies on the regulatory impact of FTO expression on adipogenesis have utilized short hairpin RNAs (shRNAs) or silencing RNAs (siRNAs) to inhibit or deplete *FTO* [61,85,86,91,92,93,94,95,96,97,98,102,110,111].

miRNA-22-3p is enriched in human milk exosomes [199,200] and is excessively overexpressed together with miRNA-148a-3p in milk exosomes of mothers delivering preterm infants [201]. Importantly, miRNA-22-3p targets *FTO* mRNA and suppresses *FTO* expression [202]. Bone marrow stem cell-derived extracellular vesicles delivering miRNA-22-3p have been shown to inhibit the MYC/PI3K/AKT pathway, thereby promoting osteogenic differentiation via *FTO* repression [202]. FTO deficiency promotes thermogenesis and white-to-beige adipocyte transition via enhanced protein expression of HIF-1α [100,101,102]. Furthermore, miRNA-22-3p via targeting hypoxia-inducible factor 1-α inhibitor (*HIF1AN*) enhances HIF-1α-stimulated glycolysis required for thermogenesis in BAT [200]. On the other hand, miRNA-22-3p directly targets CCAAT/enhancer-binding protein δ (*CEBPD*) [203], a key transcription factor involved in MCE and gene expression in early stages of white adipocyte differentiation [204] (Figure 2). It was found that circMAPK9 competes for binding miRNA-1322 in the cytoplasm, weakening the inhibitory effect of miRNA-1322 on *FTO* thereby promoting adipogenesis [205], a further observation supporting the regulatory impact of the miRNA-adjusted *FTO* expression in adipogenesis.

Recent evidence suggests that the composition of miRNAs in maternal milk is altered in women with overweight/obesity [206]. Specifically, the level of miRNA-30b in colostrum was negatively correlated with maternal pre-pregnancy BMI (*p* < 0.01) [207]. In accordance, the abundance of miRNA-30b levels was reduced in milk exosomes of mothers with obesity [208] and gestational diabetes [209]. The Targetscan Human 8.0 database [210] indicates that several abundant miRNAs found in human milk exosomes, such as miRNA-30-5p, miRNA-21-5p, and miRNA-155-5p, all target *FTO*. Notably, *MIR21* is one of the genes that displays significant changes in DNA methylation due to exclusive BF [38].

Interestingly, it has been shown in zebrafish that miRNA-30b is involved in the regulation of lipid metabolism by directly targeting *FTO* expression with subsequent FTO-mediated m6A RNA demethylation of lipogenic genes [211] (Figure 3).

Overexpression of FTO results in fat accumulation, increase of triacylglycerol and total cholesterol levels, and upregulation of PPAR-γ and C/EBPα, associated with decreased global m6A levels. Conversely, knock-down of *FTO* causes an anti-lipogenic effect, decreases triacylglycerol and total cholesterol levels, and decreases C/EBPα and PPAR-γ expression [211].

In mothers with obesity, milk exosomal miRNA-30b negatively correlated with weight, percent body fat, fat mass, and fat free mass in 6-month outcomes, especially in the exclusively BF group of infants [203]. Interestingly, the offspring of obese female Sprague Dawley rats at weaning exhibited increased hypothalamic *FTO* expression associated with increases of visceral and epididymal fat mass [212]. Unfortunately, Shah et al. [208] did not provide data on the relative distribution of WAT and BAT. There is substantial evidence in mouse models that overexpression of miRNA-30b/c induces mitochondrial and thermogenic genes, including uncoupling protein 1 (*UCP1*) and cell death-inducing DFFA-like effector A (*CIDEA*), as well as mitochondrial respiration in BAT and brite adipose tissue. This is achieved by targeting receptor-interacting protein 140 (RIP140; *NRIP1*) [213]. In contrast to physiological WAT and in vitro-differentiated wild-type adipocytes, RIP140-null cells exhibit elevated energy expenditure and express high UCP-1 levels [214]. Mice lacking the nuclear corepressor protein RIP140 are lean, demonstrate resistance to HFD-induced obesity and hepatic steatosis, and show increased oxygen consumption [215]. Notably, the correlation between miRNA-30b and brown thermogenesis was also observed in fish oil-fed C57BL/6 mice [216]. In accordance, FTO deficiency has been demonstrated to promote thermogenesis and the transition of white adipocytes to beige adipocytes through YTHDC2-mediated translation and increased HIF-1α protein expression [100]. This affects gene and miRNA expression, regulating brown adipogenesis and the browning of WAT in mice [101]. FTO deficiency has also been reported to increase *UCP1* expression, leading to increased mitochondrial uncoupling and energy expenditure, ultimately promoting the development of a brown adipocyte phenotype [102]. Additionally, FTO deficiency results in decreased expression of *FOXO1* [110], which in turn reduces the suppression of the *UCP1* promoter by FoxO1, ultimately leading to the upregulation of *UCP1* [217,218,219].

Milk exosomal miRNA-30b, miRNA-22, and miRNA-21 and other milk miRNAs directly target *FTO* [210]. This posttranscriptional adjustment may regulate the final expression level of *FTO* for tuning white and brown adipogenesis. This epigenetic regulatory system is absent in artificial formula (Figure 4).

### 3.7. Hypothalamic FTO Expression Enhances Energy Intake

FTO is highly expressed in the hypothalamus, where it controls energy intake in response to various nutritional conditions [220,221,222]. Importantly, total energy intake at 3, 6, 9, and 12 months averaged 0.36, 0.34, 0.35, and 0.38 MJ/kg/day (85.9, 80.1, 83.6, and 89.8 kcal/kg/day) among BF infants vs. 0.41, 0.40, 0.39, and 0.41 MJ/kg/day (98.7, 94.7, 93.6, and 98.0 kcal/kg/day) among FF infants, respectively. Protein intake was 66-70% higher in the FF than in the BF group during the first 6 months. Differences in energy and protein intakes were significant at 3, 6, and 9 months [223].

Johansson [224] observed that oral leucine uptake enhances FTO expression in the hypothalamic tissue of mice and regulates energy balance as well as feeding reward. Cheung et al. [131] reported that *FTO* mRNA and FTO protein levels are substantially suppressed by total AA deprivation in mouse hypothalamic N46 cells. As shown in mice, enhanced FTO expression increases fat mass and obesity via hyperphagia [225]. It is generally accepted that genetic variations of *FTO* robustly correlate with obesity, increased calorie and food intake, preferential consumption of energy-dense food, and loss of control over eating [226,227,228,229,230,231]. Recent evidence indicates that epigenetic upregulation of hypothalamic FTO expression potentially correlates with valproate-induced weight gain [232]. HFD has also been reported to upregulate hypothalamic FTO content [233]. Furthermore, hypothalamic FTO promotes HFD-induced leptin resistance in mice through increasing chemokine (C-X3-C motif) ligand 1 (CX3CL1) expression [231]. Notably, *CX3CL1* RNA is m6A-demethylated in an FTO-dependent manner. Additionally, upregulation of FTO/CX3CL1 and suppressor of cytokine signaling 3 (SOCS3) in the hypothalamus impairs leptin- and signal transducer and activator of transcription 3 (STAT3) signaling, resulting in leptin resistance and obesity [234]. Compared to wild-type (WT) mice, FTO deficiency in leptin receptor-expressing hypothalamic neurons significantly suppressed the upregulation of CX3CL1 and SOCS3, ameliorating leptin resistance under HFD [234].

Overexpression of FTO in the hypothalamic arcuate nucleus (ARC) of female rats decreased m6A RNA methylation of phospholipase C β3 (*PLCB3*), an important enzyme in the Ca^2+^ signaling pathway [123]. In the hypothalamus, FTO-mediated m6A demethylation upregulates the expression of gonadotropin-releasing hormone (GnRH) [123]. Of note, the GnRH receptor is expressed on human preadipocytes, and a GnRH agonist stimulates preadipocyte proliferation and lipid accumulation [235]. Conversely, deficient GnRH-GnRHR signaling decreased subcutaneous inguinal fat pad weight and increased glucose concentrations, potentially leading to insulin resistance in female mice [236].

Karra et al. [73] reported a connection between FTO, ghrelin expression, and disturbed brain–food cue responsivity. In adiposity-matched, normal-weight humans, they demonstrated that carriers of the *FTO* “obesity-risk” rs9939609 A allele exhibit dysregulated plasma levels of the orexigenic hormone acyl-ghrelin and reduced postprandial appetite suppression. Of note, *FTO* overexpression decreased ghrelin m6A mRNA methylation, enhancing ghrelin (*GHRL*) mRNA and peptide levels, as shown in cell models. Furthermore, peripheral blood cells from *FTO* AA individuals exhibited elevated *FTO* mRNA, reduced ghrelin mRNA m6A methylation, and increased ghrelin mRNA levels compared to TT individuals [73]. Benedict et al. [237] confirmed the correlation between FTO and ghrelin expression and lower serum concentrations of the satiety hormone leptin in older adults. Within the first 4 months of life, FF infants, as opposed to BF infants, exhibited higher ghrelin concentrations (2654.86 vs. 2132.96 pg/mL; *p* < 0.032) and elevated IGF-1 concentrations (3.73 vs. 3.15 ng/mL; *p* = 0.00) but lower serum leptin concentrations (0.68 vs. 1.16 ng/mL; *p* < 0.04) [238]. Ghrelin is implicated in the pathogenesis of diabetes and obesity, two key features of the metabolic syndrome [239,240,241,242]. Acylation of ghrelin by ghrelin-O-acyl-transferase (GOAT) is necessary for binding to and activating its receptor, the growth hormone secretagogue receptor 1a, in the hypothalamus as well as ghrelin’s orexigenic function [243,244,245]. Through an mTORC1-dependent mechanism, ETS variant transcription factor 5 (ETV5) stimulates the *GOAT* promoter and enhances GOAT expression [244]. Rats treated with *GOAT* antisense gained less weight and reduced their caloric efficiency when consuming a HFD compared to control animals, indicating that central GOAT plays a role in regulating metabolism in rats [246].

Notably, *ETV5* mRNA is a direct target of miRNA-148a-3p [247] and miRNA-200c-3p [248], two abundant miRNAs found in human milk and milk exosomes [249,250]. It is conceivable that milk exosomal miRNAs, such as miRNA-30b-5p and miRNA-22-3p, targeting hypothalamic *FTO*, as well miRNA-148a-3p and miRNA-200c-3p, targeting *ETV5*, may regulate central ghrelin action and satiety circuits. These potential satiety regulatory circuits may be controlled by breastmilk miRNAs, which however are absent in formula.

The mediobasal hypothalamus is an important region of the brain that controls both caloric intake and lipolysis [251]. In postnatal life, Iroquois homeobox protein 3 (IRX3) is highly expressed in the hypothalamus, predominantly in POMC neurons [252]. IRX3 plays a crucial role in regulating whole-body energy homeostasis [252,253]. Partial inhibition of hypothalamic IRX3 leads to increased diet-induced body mass gain and adiposity, as a result of increased caloric intake and decreased energy expenditure [252]. Notably, the long-range enhancer of *FTO* reduces hypothalamic IRX3 levels and suppresses lipolysis in peripheral adipocytes [253]. Risk alleles of the *FTO* gene are associated with reduced IRX3 expression. Inhibiting hypothalamic IRX3 reduces thermogenesis and increases body weight and WAT mass in both mice and humans [253]. Specific subsets of hypothalamic neurons also suppress lipolysis by modulating FTO expression [51].

CUX1 interacts with the binding site of *FTO* intron 1, upregulating the adjacent *RPGRIP1L* to inhibit leptin receptor transport and leptin signaling [52,53,54]. *RPGRIP1L* is located more than 100 base pairs away in the opposite transcriptional orientation to *FTO*. It facilitates leptin receptor aggregation, which regulates food intake and leptin sensitivity at the cellular level [53]. Increased FTO activity in the hypothalamus promotes leptin resistance, increasing caloric intake and reducing lipolysis [55]. CUX1 upregulates the expression of *RPGRIP1L* by binding to intron 1 of *FTO*. This interaction promotes a decrease in leptin sensitivity, ultimately promoting obesity by reducing lipolysis. Notably, CUX1 is one of the genes that shows significant CpG DNA methylation changes during infancy compared to other stages of childhood growth in girls, linking exclusive BF to longitudinal BMI changes [38].

### 3.8. FTO Expression, Circadian Rhythm, and Obesity

Feeding patterns between BF infants and FF infants differ. FF infants exhibit significantly higher feeding volumes starting from the 6th week of life onwards [254]. The human fetus during pregnancy receives timed signals from the mother’s circadian rhythms of temperature, metabolites, and hormones. After birth, the infant’s circadian rhythms are still immature [255]. Apart from external environmental cues, infants also receive maternal time signals through breastmilk, maintaining a connection to the maternal suprachiasmatic nucleus [256]. In contrast to daytime milk, night milk contains higher levels of tryptophan (TRP). These levels vary based on the newborn’s feeding schedule, which typically occurs every 3 h, regardless of the time of day [257]. A temporal relationship was observed between the circadian rhythm of 6-sulfatoxy-melatonin of the exclusive BF babies and that of TRP in the mother’s milk. Apparently TRP acts as a zeitgeber entrainment of the biological rhythms in the BF infant [258]. Lodemore et al. [259] observed that circadian rhythmicity in body temperature appeared earlier in BF compared to FF infants. Thus, the newborn receives the mother’s timed cues for the establishment of circadian clock development through breastmilk, an important argument to rely on BF [260].

Evidence accumulated over the last decade indicates that the pace of the circadian clock is controlled by m6A-dependent RNA processing [261]. m6A methylation-regulated RNA processing influences the duration and stability of the mammalian circadian clockwork [261,262]. For instance, circadian clock regulation of hepatic lipid metabolism is modulated by m6A mRNA methylation [263]. Suppression of m6A methylation and its impact on RNA processing are sufficient to slow down the circadian clock [261,264].

The m6A RNA demethylase FTO is involved in the modulation of circadian rhythms. Overexpression of FTO inhibits key regulators of the circadian clock network: circadian locomotor output cycles kaput (*CLOCK*)- and brain and muscle ARNT-like protein 1 (*BMAL1*)-induced transcription [265]. Knockout of *BMAL1* in the mouse liver disrupts lipid metabolism, exhibiting a disturbed crosstalk between the circadian gene network, mRNA m6A modifications, and metabolic state [263,266]. Cryptochrome 1 and 2 (CRY1 and CRY2) overexpression suppresses FTO and increases m6A levels in murine hippocampal neuronal cells [267]. Notably, Chong et al. [268] reported differences in the circadian feeding pattern of 12-month-old infants in relation to BF vs. FF. Compared to pre-midnight feeders, post-midnight feeders, who consumed primarily formula milk during the post-midnight period, exhibited higher daily energy, carbohydrate, fat, and protein intakes than pre-midnight feeders. Importantly, exclusive BF during the first 6 months of life was negatively associated with post-midnight feeding at 12 months [268]. Thus, BF in contrast to FF modifies the circadian pattern and lowers the amount of energy intake, potentially involving FTO-mediated changes in the circadian clock network. Feeding time and dietary nutrients are key environmental zeitgebers affecting the circadian rhythm–lipid metabolism interplay and critical factors influencing mechanisms in obesity development [269]. Feeding time is considered a potential external synchronization cue that fine-tunes the timing of the circadian rhythms in metabolic peripheral clocks [270]. Disruption of physiological circadian rhythms is a potential negative effect of FF [268], which may have long-term negative consequences for metabolic health. Notably, a disturbed interplay between the circadian clock and lipid metabolism plays a critical role in the development of obesity [269,270,271,272,273].

Infants who consumed TRP-enriched formula at night showed improved sleep parameters compared to those who consumed it during the day [274]. However, TRP serves not only as a precursor of melatonin but also of nicotinamide adenine dinucleotide (NAD), phosphorylated NAD (NADP), and NADPH [275,276,277]. Recent evidence indicates that NADP and NADPH reduce RNA m6A methylation, increasing adipogenesis by direct binding to the FTO protein and enhancing its enzymatic activity [93]. The TRP concentrations in high-protein formula (2.7 g/100 kcal) [127] exceed those in low-protein formula (1.65 g/100 kcal) [125] and mature breastmilk [278], potentially explaining the accelerated postnatal weight gain in infants and increased risk of obesity [128]. FTO likely plays a role in the conversion of TRP to kynurenine (KYN) [279]. The KYN pathway is activated in human obesity and shifts towards KYN monooxygenase activation [280]. KYN levels have been positively associated with BMI and a higher HOMA2-IR insulin resistance index [280]. Marszalek-Grabska et al. [281] recently reported that KYN is also a component of human milk and is present in commercial baby formulas in amounts that noticeably exceed its physiological range.

Of note, KYN is a precursor of NADP [282], which activates FTO [93]. Huang et al. [283] provided evidence that BCAAs decrease the concentration of NADPH in adipose tissue and 3T3-L1 cells by reducing glucose-6-phosphate dehydrogenase (G6PD) expression. The reduced NADPH attenuates the expression of FTO, leading to an increase in m6A levels of *CCNA2* and *CDK2* mRNA, which prevents MCE of preadipocytes by reducing the expression of cyclin A2 (*CCNA2*) and cyclin-dependent kinase 2 (*CDK2*). High m6A levels of *CCNA2* and *CDK2* mRNA are recognized by YTH N6-methyladenosine RNA binding protein 2 (YTHDF2), resulting in mRNA decay and a reduction in their protein expressions [283].

Intracellular nicotinamide phosphoribosyltransferase (NAMPT), also known as visfatin, catalyzes the rate-limiting step in the NAD+ biosynthesis pathway from nicotinamide [284,285,286,287]. Remarkably, homozygous newborns for the at-risk allele (AA) of rs9939609 SNP in *FTO* had 37% higher cord blood visfatin concentration and 17%, 20%, and 17% higher total, truncal, and abdominal fat mass at age 2 weeks, respectively, than newborns carrying the T allele [288]. These findings point to an early interactive involvement of FTO and visfatin in fat accretion and abdominal fat mass development in human neonates [288].

Remarkably, CLOCK:BMAL1 controls the circadian expression of NAMPT, which contributes to the circadian rhythm of several cell functions [289,290]. Studies in mice using genetically modified models have shown that NAMPT plays a critical role in lipogenesis and lipolysis, specifically in adipocytes [291]. Deletion of NAMPT in adipocytes resulted in a disrupted circadian transcriptome in WAT and BAT, with BAT showing a disrupted rhythm of metabolites in energy metabolism [292].

Increased FTO-mTORC1-S6K1 signaling may also impair the early expansion of beige adipose tissue and BAT via nuclear S6K1 phosphorylation of BMAL1, allowing the recruitment of EZH2 and suppressing *ADIPOQ* expression [172]. Notably, adiponectin plays an important role for Sca1 + CD34--adipocyte precursor cells, associated with hyperplastic expansion and beiging of white adipocytes [173]. Thus, FTO-mTORC1-S6K1 controls the nuclear traffic of BMAL1, critically involved in the regulation of the circadian clock as well as adipocyte fate decisions.

Recommendations for optimized practices for FF to reduce excess or rapid weight gain include providing formula with lower protein content, avoiding adding cereals to bottles, not putting a baby to bed with a bottle, and not overfeeding formula [293]. While formula bottles given at nighttime with increased protein (TRP) and KYN content may improve the infant’s sleeping time, they may simultaneously increase the risk of TRP-KYN-NAMPT-NADP-induced FTO overexpression, promoting weight gain and adipogenesis.

### 3.9. Potential Impact of Exosomal Milk miRNAs in FTO-Circadian Clock Regulation

Recent evidence indicates that extracellular vesicles participate in circadian rhythm synchronization [294]. There is a growing attention given to the crucial involvement of miRNAs in regulating various settings of circadian clock function [295,296,297]. For instance, miRNA-17-5p and miRNA-29b-3p directly regulate circadian gene expression in the maturing islet cells of 10-day-old postnatal rats [298]. In contrast to exosome- and miRNA-deficient infant formula [186,187,193], human milk and milk exosomes deliver highly conserved miRNAs such as miRNA-148a-3p and miRNA-30b-5p to the infant [180,200,298,299,300,301,302]. miRNA-148a-3p and miRNA-30-5p target peroxisome proliferator-activated receptor gamma co-activator-1 alpha (*PPARGC1A*) [303,304,305], a co-transcription factor promoting the expression of *BMAL1* [306].

Breastmilk contains miRNA-17-5p [307], which is highly upregulated in preterm colostrum [294] and represses *CLOCK* gene expression [308]. miRNA-17-5p equilibrates the period length of the circadian rhythm, potentially via the modulation of the CRY1 level through CLOCK in the suprachiasmic nuclei. Both an increase and decrease in miRNA-17-5p levels in the suprachiasmatic nucleus of mice result in higher CRY1 levels and a shorter free-running period [305].

FTO binds to CRY1 and CRY2 in a circadian manner [263]. Additionally, FTO itself is a target gene of miRNA-22-3p, miRNA-30b-5p, miRNA-21-5p, and miRNA-155-5p [196,207,210,309]. These miRNAs are abundant in human colostrum and mature milk [310]. Unlike milk miRNA-deficient formula, breastmilk-derived exosomal miRNAs can target crucial genes of the circadian clock, including *FTO*. This may play a critical role in adjusting the neonatal circadian rhythm and food/energy intake to prevent the development of obesity. A significant role for mTORC1 in circadian timekeeping and in linking metabolic states to circadian clock functions has recently been reported [311].

### 3.10. Zinc Finger Protein 217 Promotes FTO Expression and Adipogenesis

Zinc finger protein 217 (ZFP217) is a member of the kruppel-type zinc finger protein family and exerts various functions in mammalian cell differentiation and development. Recent evidence identifies its physiological roles in adipogenesis [312] and the control of m6A deposition in embryonic stem cells [313]. ZFP217 regulates adipogenesis by controlling MCE in a METTL3-m6A-dependent manner, and *ZNF217* knockdown inhibits adipogenesis [314]. Recent evidence also shows that ZFP217 modifies m6A mRNA methylation by activating the transcription of *FTO* [92]. Depletion of ZFP217 compromises the adipogenic differentiation of 3T3L1 cells and results in a global increase of m6A modification. Furthermore, ZFP217 interacts with YTHDF2, which is critical for allowing FTO to maintain its interaction with m6A sites on various mRNAs. The loss of ZFP217 deceases FTO and enhances m6A levels [92]. *Zfp217*^+/−^ mice challenged with a HFD exhibit less weight gain than *Zfp217*^+/+^ mice. Histological analyses show that *Zfp217*^+/−^ mice fed an HFD had much smaller white adipocytes in inguinal WAT. These mice had improved metabolic profiles, including improved glucose tolerance and insulin sensitivity as well as enhanced energy expenditure compared to *Zfp217*+/+ mice under an HFD [314]. In contrast, HFD-fed *Zfp217*^+/+^ mice exhibited increased adipogenesis-related genes and decreased metabolic thermogenesis-related genes in inguinal WAT as compared to *Zfp217*^+/−^ mice [314]. Moreover, adipogenesis was markedly reduced in mouse embryonic fibroblasts from *Zfp217*-deleted mice [315]. These findings indicate that ZFP217 is a key regulator of FTO expression, critically involved in adipogenesis and energy metabolism.

Three miRNAs (miRNA-503-5p, miRNA-135a-5p, and miRNA-19a-3p) that target *ZNF217* were found to suppress the process of adipogenesis [312]. This finding clearly demonstrates that miRNAs are able to control ZNF217-regulated adipogenesis. Notably, *ZNF217* is also a direct target of miRNA-148a-3p, miRNA-200bc-3p, and miRNA-17-5p [316], which are abundantly expressed miRNAs in human milk and milk exosomes. Thus, milk exosomal miRNAs may also be involved in calibrating ZFP217-stimulated *FTO* expression, a potential posttranscriptional regulatory network of BF that is missing in FF. In addition, ZFP217/FTO-induced m6A RNA demethylation may regulate the expression and function of circadian clock genes.

### 3.11. FTO Deficiency and Long Non-Coding RNA H19 Promote Brown Adipose Tissue

Over the last decade, accumulated evidence underlines that activating BAT and browning of WAT can protect against obesity and obesity-related diseases [317]. BAT plays a key role in energy metabolism and thermoregulation during the perinatal period in all mammals [318]. The thermogenic activity of BAT is significant for maintaining a sufficiently high core body temperature in human infants [319]. Exclusive BF can help maintain the newborn’s body temperature in a physiological range [319]. Whereas 100% of exclusive BF infants exhibited a normal body temperature 0–72 h after birth, mixed feeding showed hypothermia in 22% of infants [320].

Recent evidence indicates that the expression of lncRNA H19 is increased after METTL3 and METTL14 binding to it, resulting in m6A-mediated methylation. Knocking down these proteins inhibits H19 expression [321]. In humans, an inverse association between H19 expression and BMI was observed. Overexpression of H19 enhanced, while H19 silencing impaired adipogenesis, oxidative metabolism, and mitochondrial respiration in BAT but not in WAT adipocytes [322]. LncRNA H19 protects against dietary obesity by constraining the expression of monoallelic genes in BAT. Loss of H19 in brown fat reduces energy expenditure and sensitizes towards obesity [322].

Increased expression of the FTO may decrease H19 expression, impairing BAT metabolism and thermogenesis. Conversely, FTO deficiency has been related to BAT differentiation and thermogenesis [100,101,102]. Specifically, FTO deficiency promotes thermogenesis and the transition of white-to-beige adipocytes through YTHDC2-mediated translation, enhancing protein expression of HIF-1α [100]. It also affects the gene and miRNA expression involved in brown adipogenesis and browning of WAT in mice [101] and upregulates uncoupling protein 1 (UCP-1), thereby enhancing mitochondrial uncoupling and energy expenditure, inducing a brown adipocyte phenotype [102] and potentially upregulating H19 expression.

Elevated expression of the lncRNA H19, which sponges let-7 miRNA [323], may activate the expression of HIF-1α [324]. HIF-1α, in turn, activates the transcription of thermogenic genes that promote UCP-1 expression and the browning process [100]. A preliminary pilot study suggests that human milk exosomes contain lncRNA H19 (R.W. and B.C.M, unpublished), which could potentially provide exosomal support for BAT development, which is lacking in exosome-deficient infant formula.

## 4. Discussion

After decades of experimentation to improve FF, growth patterns and body composition development still differ between FF infants and BF infants, obviously contributing to an increased risk of obesity among FF infants [325]. Excessive protein and mismatched AA intake by artificial FF in comparison to nature’s gold standard, i.e., exosome-delivering human milk, is a matter of great concern [126,127,325]. It may never be possible to avoid protein excess with FF, as parents will not always comply with recommended formula dosing, especially when providing a protein-enriched night bottle to increase the infant’s sleep period. The protein content and dynamics of protein intake are not the only critical determinants to adjust appropriate FTO signaling during the postnatal period of metabolic programming of the infant. The levels of KYN, a downstream metabolite of TRP and an upstream precursor of NADP [326], are significantly higher in formula and may thus further increase NADP-stimulated FTO activation. Remarkably, TRP and KYN are positively associated with the FTO rs9939609 A allele [279]. Obese adults exhibit marked elevations of KYN serum levels and a significantly elevated KYN/TRP ratio [327,328,329]. In mice, adipocyte-derived KYN promotes obesity and insulin resistance by activating the arylhydrocarbon receptor (AhR)/STAT3/IL-6 signaling pathway [330]. Recent evidence indicates that melatonin, another downstream metabolite of TRP [273,274,275,276,277,278], stimulates the expression of miRNA-148a-3p [331], a potential link between TRP metabolism and miRNA signaling. Currently, there is no information regarding exosomal miRNA expression in human milk at night time. Oxytocin, a key driver of lactation [332], has been shown to enhance the expression of miRNA-148a-3p, miRNA-30b-5p, and miRNA-21-5p in colostral milk exosomes [333], pointing to the close hormonal interaction between milk production and exosomal miRNA content missing in artificial formula.

Apparently, FF at multiple regulatory steps fails to adjust the appropriate magnitude of FTO signaling. Recent evidence underlines that BF delivers a highly sophisticated vesicular system that transmits a complex spectrum of RNAs critically involved in the epigenetic programming of the infant [178,179,180,181,182,183,184]. It is conceivable that not only deviated protein signaling but also the deficiency of milk exosomal miRNAs in formula enhances the risk of FTO overexpression, leading to obesigenic deviations in the RNA methylome.

Unfortunately, a century ago, FF was developed under the simplified view that milk is “just food” [334]. Today, we have compelling evidence that FF leads to significant deviations in DNA and most likely RNA methylation patterns. Accumulated evidence provided in this review demonstrates that FTO modifies the RNA methylome and induces the expression of multiple cell cycle and lipogenic genes (Table 2) involved in the development of WAT but suppresses genes important for the generation of BAT and thermogenesis (Table 1). The appropriate calibration of FTO activity and the extent of m6A RNA demethylation play pivotal roles in the process of adipogenesis of WAT, BAT development, and the regulation of circadian rhythm and energy intake. Obviously, BF and FF significantly differ with regard to their impact on epigenetic regulation. Adverse effects of FF on the m6A RNA methylome are thus a new matter of concern, as m6A RNA demethylations are critically involved in gene expression, tissue development, and adipogenesis [50,62]. There is common consent that nutrition during the first 1000 days is of critical importance for infant development and the prevention of obesity [335,336]. DNA methylation mediates not only the association between BF and early-life growth trajectories [33,34,35,36,37,38,39,40] but apparently also the appropriate regulation of the RNA methylome. Already, a decade ago, early postnatal life was identified as a critical time window for the determination of long-term metabolic health [28]. High-protein formula-induces epigenetic modifications of FTO, which apparently increases the number of adipocytes during a critical developmental window. During lifetime challenges with HFD, those infants adversely primed with developmental deviations of their adipose tissue may have a higher disposition towards obesity compared to BF infants carrying a normally developed adipose tissue during the postnatal period. Notably, the gain-of-function FTO rs9939609 variant is linked to a higher BMI and increase of total energy intake, whereas lower dietary protein intake can weaken the connection between the *FTO* genotype and adiposity in children and adolescents [177]. Exclusive BF acts antagonistically to the *FTO* rs9939609 risk allele by the age of 15 years in boys and girls [76]. These findings clearly emphasize the importance of adequate FTO signaling during the postnatal developmental window of adipogenesis.

Figure 5 summarizes all potential epigenetic m6A-dependent metabolic deviations mediated by high-protein FF-induced FTO overactivation, including deficiencies in critical miRNAs that would be able to silence and thus balance FTO expression.

Although we now know that FF does not guarantee proper physiological postnatal epigenetic programming during a critical and vulnerable period in our infants’ lives, sales of infant formula are rising dramatically worldwide, a fact heavily criticized by the authors of the Lancet Breastfeeding Group [337,338]. In particular, infants carrying the FTO rs9939609 risk allele exhibit an enhanced risk for FTO-mediated adipogenic signaling by FF, whereas prolonged exclusive BF reduces the risk of BMI increase in childhood [76]. In this regard, it is also noteworthy to mention that exposure of pregnant mice to bisphenol A leads to obesity in the F2 progeny due to increased food intake, with an epiphenotype that can be transmitted up to the F6 generation. Analysis of chromatin accessibility in the sperm of the F1-F6 generations exhibits alterations at sites containing binding motifs for CTCF at two cis-regulatory elements of the *FTO* gene that correlate with the transmission of obesity [146]. These findings suggest that not only genetic variants but also epigenetic externally introduced changes in *FTO* expression during the perinatal period can result in the same phenotypes and can be transmitted over generations [147]. Placental FTO expression also correlates with fetal growth [339]. After birth, the secretory products of the mammary gland take over FTO-mediated growth control. Human milk, the masterpiece of mammalian lactation that has evolved over millions of years [340,341], ensures optimized epigenetic signaling at the DNA and RNA levels for the appropriate postnatal metabolic programming of the infant. Of course, the feeding pattern after the lactation period as well as the time point and type of introduction of processed foods will modify epigenetics and later disease development. Nevertheless, the lactation period is a vulnerable time window of postnatal tissue development that requires optimal protection.

Changes of the intestinal microbiome might also modify FTO signaling. The milk of mothers giving birth via cesarean section contains lower amounts of miRNA-148a-3p compared to mothers giving birth by vaginal delivery [195]. FTO-deficient mice exhibit a specific bacterial signature, characterized by a higher abundance of *Lactobacillus* and suppression of inflammation [342]. Preliminary evidence indicates that the gut microbiota might modulate responses of lipid metabolism to nutritional intervention in individuals with different *FTO* genotypes [343]. It has been shown in C57BL/6 mice that dietary bovine milk exosomes elicit beneficial changes in bacterial communities [344]. Intestinal dysbiosis in preterm infants often precedes necrotizing enterocolitis [345], whereas milk exosomes and milk exosomal miRNAs including miRNA-148a-3p [191], lncRNAs, and mRNAs might protect against necrotizing enterocolitis [193,346,347] and ulcerative colitis, as shown in a genetic mouse model [348]. Thus, milk exosome–gut microbiome–FTO interactions appear as promising areas for future research.

Limitations of this review are the fact that statements on the impact of FTO signaling are based on correlative, epidemiological, and translational studies in humans and animal models and not on molecular data obtained from adipose tissues of FF versus BF infants, which is not suitable for ethical reasons. On the other hand, infant formula was introduced without providing any epigenetic tissue analyses of FF versus BF infants prior to marketing.

## 5. Conclusions

Experimental and translational evidence points to enhanced FTO signaling by FF, which may disturb m6A-dependent RNA regulation in early adipogenesis and lipogenesis. When comparing the differences of BMI trajectories of BF vs. FF, variations in total protein content and AA composition are not the only critical variables: Total protein intake as well as the relative amounts of TRP, KYN, BCAAs, and exosomal miRNAs may synergistically control the final magnitude of FTO expression and activation and, consequently, downstream FTO-mediated m6A-dependent RNA translation. Accumulating evidence supports the view that FF leads to aberrant early fate decisions in adipose stem cell homeostasis, promoting excessive preadipocyte linage commitment and preadipocyte cell mass expansion via increased FTO/mTORC1/S6K1 signaling, as well as absent milk miRNAs, disturbing the suppressive effects WNT signaling on postnatal adipogenesis. It may be relevant to include primate and human adipose tissue-specific expression of *FTO* for experimental evaluation of the effects of human milk versus formula, which of course is a challenge due to ethical limitations.

Infant formula is still and may always be in an “experimental stage”, which is not equivalent to the epigenetic and metabolic effects of exclusive BF on physiological stem cell differentiation. The historical misconception regarding “milk is just food” may be a critically overlooked error promoting our ongoing obesity epidemic. From a public health perspective, it is high time to switch back from bottle to breast to prevent postnatal obesigenic programming of our infants of all mothers who are able to feed and program their newborns by BF. It is questionable whether future formula improvements may ever reach the quality, quantity, and dynamics of epigenetic regulations guaranteed by natural BF.

## Figures and Tables

**Figure 1 nutrients-16-02451-f001:**
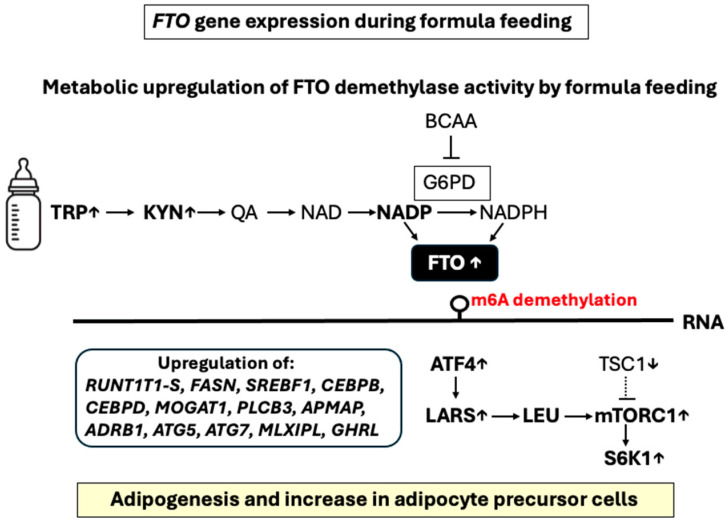
Metabolic activation of the FTO protein through FF. FF leads to excessive intake of tryptophan (TRP) and kynurenine (KYN), which are then converted to quinolinic acid (QA) and nicotinamide adenine dinucleotide (NAD). NAD is further metabolized into nicotinamide adenine dinucleotide phosphate (NADP) and NADPH. Both NADP and NADPH bind to the FTO protein, increasing its enzymatic activity. However, the conversion of NADP to NADPH can be inhibited by high levels of branched-chain amino acids (BCAA) through the inhibition of glucose-6 phosphate dehydrogenase (G6PD). The activated FTO upregulates the expression of multiple lipogenic genes and ATF4 through m6A RNA demethylation. ATF4, in turn, upregulates the leucyl-tRNA synthetase (LARS), thereby linking leucine (LEU) levels to mTORC1 activation. FTO-mediated suppression of tuberous sclerosis complex 1 (TSC1) destabilizes the TSC1/TSC2 complex, enhancing insulin/IGF-1/PI3K/AKT signaling towards mTORC1 and its downstream target S6 kinase 1 (S6K1), which finally enhances the number of adipocyte precursor cells.

**Figure 2 nutrients-16-02451-f002:**
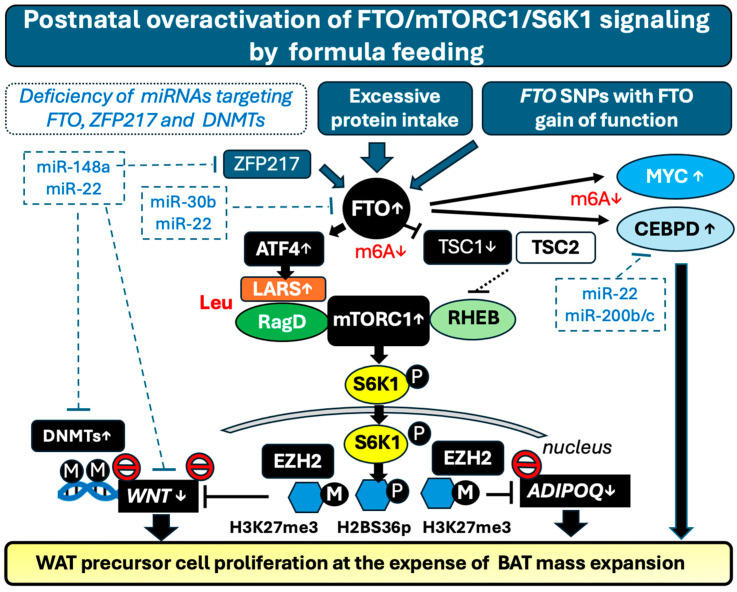
Involvement of overactivated FTO signaling promotes wingless (*WNT*) gene suppression-mediated adipocyte precursor cell proliferation. FTO expression is upregulated by excessive milk protein intake, deficiency of milk exosomal miRNAs targeting FTO, as well as FTO gain of function SNPs. FTO-mediated upregulation of activating transcription factor 4 (ATF4) enhances the expression of leucyl-tRNA synthase (LARS). LARS interacts with Ras-related GTP-binding protein D (RagD), enhancing leucine (Leu) signaling for activation of mechanistic target of rapamycin complex 1 (mTORC1). Furthermore, FTO mediates suppression of TSC complex subunit 1 (TSC1), a component of the inhibitory TSC1/TSC2 complex, enhancing mTORC1 activation via growth factor/PI3K/AKT signaling. The downstream target of mTORC1, the phosphorylated S6 kinase 1 (S6K1-p), enters the nucleus and phosphorylates the histone protein H2B (H2BS36p), which allows the attraction of enhancer of zeste homolog 2 (EZH2), promoting the trimethylation of histone 3 (H3K27me3) and suppressing nearby *WNT* genes involved in adipocyte stem cell homeostasis. In a similar fashion, S6K1-mediated activation of EZH2 results in H3K27me3-mediated suppression of adiponectin (*ADIPOQ*) expression. Suppression of *ADIPOQ* reduces the development of beige/brown adipocytes. In addition, the absence of milk exosomal miRNAs (miRNA-148a-3p, miRNA-22-3p), that target DNA methyltransferase 1 and 3A (DNMTs), increases *WNT* promoter methylation, further suppressing *WNT* gene expression. On the other hand, miRNA-148a-3p and miRNA-22-3p directly target *WNT10B* mRNA. Enhanced FTO-mTORC1-S6K1 signaling may represent the major driving mechanism enhancing white adipocyte precursor cell proliferation via *WNT* suppression, supported by FTO-mediated activation of *MYC* and *CEBPD* expression at the expense of adiponectin-dependent beiging/browning of white adipocytes. Enhanced FTO-mTORC1-S6K1 signaling thus disturbs the relation between white adipose tissue (WAT) and brown adipose tissue (BAT) development.

**Figure 3 nutrients-16-02451-f003:**
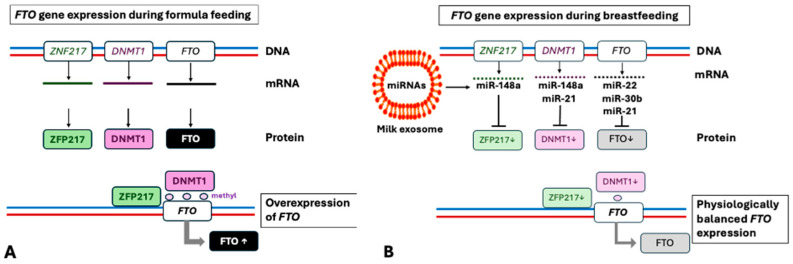
*FTO* gene expression through formula feeding (**A**) compared to breastfeeding (**B**). Artificial formula lacks milk exosomal miRNAs (miRs). The most abundant miRNAs in milk exosomes (miR-148a-3p, miR-22-3p, miR-30b-5p, miR-21-5p) target the mRNAs of zinc finger protein 217 (*ZNF217*), DNA methyltransferase 1 (*DNMT1*), and fat mass- and obesity-associated gene (*FTO*), which appear to reduce their mRNA transcription levels. Both reduced expression of DNMT1 (resulting in less *FTO* CpG methylation) and reduced expression of ZFP217 (a transcriptional enhancer of FTO), combined with miRNAs directly targeting *FTO* mRNA, ultimately decrease total FTO protein expression. The suppression of DNMT1 by miRNA-148a-3p may inhibit early adipogenic lineage commitment.

**Figure 4 nutrients-16-02451-f004:**
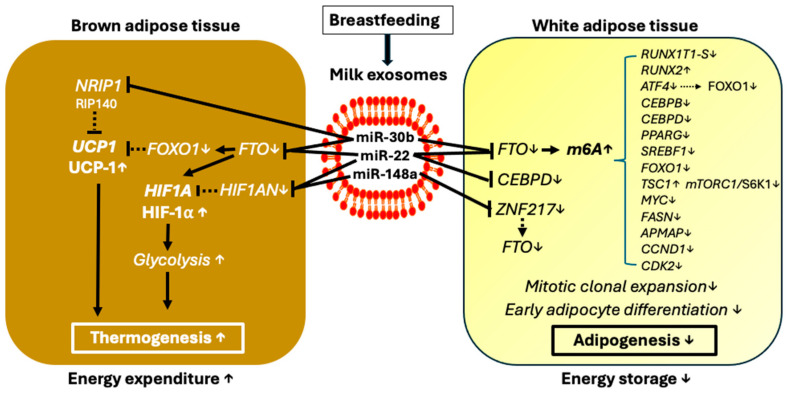
Potential impact of milk exosome-derived miRNA gene regulation of key transcription factors involved in gene expression of brown and white adipocytes. The most abundant exosomal miRNAs of human milk suppress *FTO*, enhancing the expression of uncoupling protein 1 (UCP-1) and hypoxia-inducible factor 1α(HIF-1α) and stimulating glycolysis and thermogenesis of brown adipocytes. Milk exosomal miRNA-mediated suppression of *FTO* in white adipocytes reduces the expression of genes involved in mitotic clonal expansion, early adipocyte differentiation, and lipid accumulation of white adipocytes. Excessive expression of milk exosomal miRNA-22-3p and miRNA-148a-3p under preterm birth conditions supports thermogenesis, essential for cold-sensitive preterm infants, via suppression of receptor interacting protein 140 (*NRIP1*), a negative regulator of the *UCP1* promoter, and suppression of hypoxia-inducible factor 1α inhibitor (*HIF1AN*), a negative regulator of HIF-1α and HIF-1α-induced glycolysis. Apparently, milk exosomal miRNAs balance the postnatal development and epigenetic functioning of brown and white adipose tissue, a physiological gene regulatory mechanism provided by exclusive breastfeeding but missing in artificial formula.

**Figure 5 nutrients-16-02451-f005:**
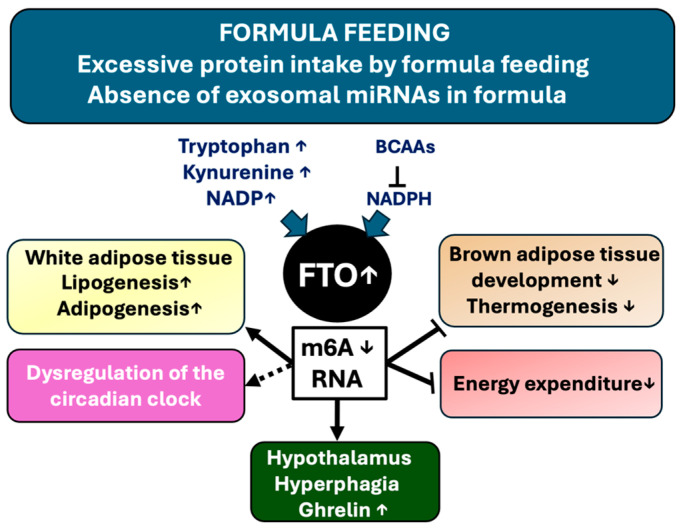
FTO overactivation by formula feeding and resulting metabolic deviations through increased m6A RNA demethylation.

**Table 1 nutrients-16-02451-t001:** The effects of FTO-dependent m6A demethylation in adipocytes, hepatocytes, skeletal muscle cells, blood mononuclear cells, and embryonic stem cells.

Cell Type	Finding	Reference
Adipocytes	Inverse correlation during adipogenesis between m6A RNA levels and *FTO* gene expression. The adipogenic role of FTO is due to the splicing of *RUNX1T1* to the short splice variant *RUNX1T1-S*	[61]
	The adipogenic effect of FTO is transmitted by mitotic clonal expansion (MCE), an early stage in adipogenesis	[85]
	The demethylase activity of FTO is required for preadipocyte differentiation	[86]
	FTO enhances catenin β1 (*CTNNB*) expression, promoting lipogenesis in a m6A demethylation-dependent manner	[87]
	FTO promotes adipogenesis and maintenance of lipid content in mature adipocytes by enabling *CEBPB*-driven transcription and expression of *CEBPD*	[88]
	FTO promotes adipocyte differentiation via β1-adrenergic receptor (*ADRB1*) gene regulation through m6A modification in Hycole rabbits	[89]
	*FTO* SNP rs1421085 increases the number of white adipocytes and lipid storage but reduces mitochondrial thermogenesis	[90]
	FTO regulates autophagy and adipogenesis	[91]
	Zinc finger protein 217 (ZFP217) stimulates adipogenesis via activation of *FTO* transcription and by posttranscriptional interaction with YTHDF2	[92]
	NADP binds to the FTO protein, increasing its activity and thereby promoting m6A demethylation and adipogenesis	[93]
	FTO promotes the expression of the adipocyte plasma membrane-associated protein (*APMAP*) gene by YTHDF2 recognition, stimulating adipocyte differentiation	[94]
	Via m6A modification, FTO upregulates MYC, a critical transcription factor promoting multipotent adipocyte stem cells to the adipogenic lineage	[95,96]
	FTO knockdown shows that increased mRNA m6A methylation downregulates porcine adipogenesis	[97]
	Epigallocatechin gallate-mediated FTO inhibition suppresses adipogenesis by inhibiting MCE in an m6A-YTHDF2-dependent manner	[98,99]
	FTO deficiency promotes thermogenesis and the transition of white-to-beige adipocytes via YTHDC2-mediated translation and increased expression of hypoxia-inducible factor-1α (*HIF1A*)	[100]
	FTO deficiency in mice modifies gene and miRNA expression involved in brown adipogenesis and browning of WAT	[101]
	FTO deficiency upregulates uncoupling protein 1 (*UCP1*) and subsequently enhances mitochondrial uncoupling and energy expenditure, resulting in the induction of the brown adipocyte phenotype	[102]
	Metformin inhibits FTO and counteracts high-fat diet (HFD)-induced obesity in mice in an m6A-YTHDF2-dependent manner	[103]
Hepatocytes	FTO enhances lipid accumulation in hepatocytes by upregulation of the nuclear translocation and maturation of SREBP1c, thereby enhancing the transcriptional activity of the lipid droplet-associated protein CIDEC	[104]
	FTO-mediated RNA demethylation reduces mitochondria numbers and stimulates hepatic fat accumulation	[105]
	Elevation of FTO stimulates liver steatosis by m6A demethylation and increases the stability of sterol regulatory element-binding transcription factor 1 (*SREBF1*) and carbohydrate response element-binding protein (*CHREBP*; *MLXIPL*) mRNAs	[106]
	FTO promotes hepatic gluconeogenesis and lipid accumulation by increasing the expression of activating transcription factor 4 (ATF4), which works in conjunction with forkhead box O1 (FoxO1)	[107]
	Glucocorticoid-mediated *FTO* transactivation enhances lipid accumulation in hepatocytes by m6A demethylation of lipogenic mRNAs	[108]
	FTO induces hepatic lipogenesis through an m6A demethylation-dependent increase of fatty acid synthase (*FASN*) mRNA	[109]
	Entacapone, a chemical FTO inhibitor, mediates metabolic regulation through FOXO1 in HepG2 cells. FTO via m6A demethylation upregulates *FOXO1* expression	[110]
Skeletal muscle cells	AMP-activated protein kinase (AMPK) stimulates lipid accumulation in skeletal muscle cells through FTO-dependent m6A demethylation	[111]
Human endothelial cells	Atorvastatin reduces FTO protein expression via a geranylgeranyl pyrophosphate–dependent and proteasome-dependent mechanism. After FTO silencing, KrüKrüüppel-like factor 2 (KLF2) transcripts with higher levels of m6A modification in their 3′ untranslated regions are captured by YTHDF3 stabilizing KLF2 mRNA, increasing KLF2 expression. KLF2 is a neg-ative regulator of PPAR-γ and adipogenesis, a potential explanation for atorvastatin-induced weight loss	[112,113,114]
Blood mononuclear cells	FTO reduces ghrelin m6A levels and subsequently enhances ghrelin (*GHRL*) mRNA abundance in *FTO* SNP rs9939609 (risk allele A) carriers	[73]
Embryonic stem cells	FTO demethylates the m6A level of long-interspersed element-1 (*LINE1*) RNA in mouse embryonic stem cells, enhancing *LINE1* RNA abundance and modifying the local chromatin state, which changes the transcription of *LINE1*-containing genes	[115]
Bone marrow mesenchymal stem cells	FTO overexpression decreases the m6A RNA and total level of runt-related transcription factor 2 (*RUNX2*) mRNA, suppressing osteogenic differentiation. Decreased expression of *RUNX2* promotes adipogenic differentiation. In contrast, intake of cow’s milk including bovine milk miRNAs increases *RUNX2* expression in peripheral blood mononuclear cells of healthy adult human volunteers	[116,117,118,119,120,121]

**Table 2 nutrients-16-02451-t002:** Selected genes regulated by FTO-mediated m6A mRNA demethylation.

Genes upregulated by FTO-mediated m6A mRNA demethylation:
Runt-related transcription factor 1, translocated to 1 short form (*RUNX1T1-S*) [61] Cyclin D1 (*CCND1*) [103,122] Cyclin-dependent kinase 2 (*CDK2*) [103] Fatty acid synthase (*FASN*) [105,109] Stearoyl-CoA desaturase (*SCD*) [105] Monoacylglycerol O-acyltransferase 1 (*MOGAT1*) [105] Fatty acid-binding protein 4 (*FABP4*) [103] Lipoprotein lipase (*LPL*) [103] Sterol regulatory element-binding transcription factor 1 (*SREBF1*) [106] Peroxisome proliferator-activated receptor-γ (*PPARG*) [103] MLX-interacting protein-like (*MLXIP*) [106] CCAAT/enhancer-binding protein alpha (*CEBPA*) [103] CCAAT/enhancer-binding protein beta (*CEBPB*) [88] CCAAT/enhancer-binding protein delta (*CEBPD*) [88] MYC protooncogene (*MYC*) [95,96] Forkhead box O1A (*FOXO1A*) [110] Activating transcription factor 4 (*ATF4*) [107] Beta-1 adrenergic receptor (*ADRB1*) [89] Catenin beta 1 (*CTNNB1*) [87] Autophagy-related 5 (*ATG5*) [91] Autophagy-related 7 (*ATG7*) [91] Ghrelin (*GHRL*) [73] Adipocyte plasma membrane-associated protein (*APMAP*) [94] Phospholipase C beta-3 (*PLCB3*) [123]
Genes upregulated by suppression of FTO-mediated m6A mRNA demethylation:
Hypoxia-inducible factor 1, alpha subunit (*HIF1A*) [99]

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
