# Peer review of "Risk of Fat Mass- and Obesity-Associated Gene-Dependent Obesogenic Programming by Formula Feeding Compared to Breastfeeding"

_nutrients, 2024, doi:10.3390/nu16152451_

Round 1

Reviewer 1 Report

Comments and Suggestions for Authors

In the manuscript “Risk of FTO m6A RNA Demethylase-Dependent Obesogenic Programming by Formula Feeding Compared to Breastfeeding”, Bodo C. Melnik, et al. screened the literature in PubMed and concluded that formula feeding may lead to hyperactivation of FTO signaling, which may interfere with m6A RNA-dependent regulation of adipogenesis and obesity. In general, in the study, the authors analyzed the importance of breastfeeding in regulating the epigenetic inheritance of infants and reducing the risk of obesity and suggested that breastfeeding should be preferred. However, there are some formatting errors in the tables and images of this article, and appropriate revisions are suggested. 

Questions were raised as below and needed to be addressed.

1. Suggest adding something about the specific nutrient differences between formula and breast milk.

2. Avoid abbreviations in the title

3. the review could be written in a different format than a research article, and the experimental methodology could be presented differently.

4. there is an irregular space in line 401.

5. 3.9 and 3.10 should read 3.9. and 3.10.

6. Pay attention to the format of the p-value, e.g. lines 184 and 185.

7. FTO should be italicized as a gene name in the first line of Figure 1.

Author Response

Response to Reviewer 1

We would like to express our gratitude to R1 for providing valuable comments and suggestions for improvement.

Specifically:

R1 identified formatting errors in Tables and Figures, which have been rectified in the revised manuscript.

Ad1) R1 suggest adding something about the specific nutrient differences between formula and breastmilk.

The differences between human milk and artificial formula are significant, and most readers in the field interested in our subject are aware of this fact. Our focus was to clarify the specific differences between FTO signaling and RNA methylation in relation to adipogenesis in breastmilk versus formula. This required a total of 344 references, so we did not delve into other nutrient differences. Instead, we concentrated on nutrients and biomolecules that are functionally linked to FTO expression and downstream signaling, such as total and specific amounts of protein, amino acids, kynurenine, exosomes, and microRNAs.

Ad 2) R1 wished to avoid abbreviations in the title.

Accordingly we modified and simplified the title of our review article.

Ad 3) R1 suggested that the review could be written in a different format than a research article, and the experimental methodology could be presented differently.

We followed the guidelines for Nutrients review articles. Our two other reviewers and ourselves were satisfied with the format of the manuscript.

Ad 4-6) R1 identified several typing errors and formatting variations in the p-values, which we have corrected in the revised version.

Ad 7) R1 requested that “FTO” be italicized as a gene name in the first line of Figure 1.

Since Figure 1 depicts aspects of FTO gene expression, we have made the necessary revisions to the figure.

Reviewer 2 Report

Comments and Suggestions for Authors

Review of Risk of FTO m6A RNA Demethylase-Dependent Obesogenic Programming by Formula Feeding Compared to Breastfeeding

 Abstract

The abstract should be improved to describe the purpose of the review, "Le but"?. The sentence in lines 21-23 does not make it clear that this (GMP) procedure was performed in the current manuscript. The phraseology could be changed to “Based on literature search through Web of Science, Google Scholar and PubMed databases concerning….”. Please consider improving this sentence.

Introduction

Line 37 – insert comma after hypothesis.

Not enough background information on FTO, epigenetics, and rationale for the review. Freedom To Operate, as in intellectual property law, may be the first nuance striking the reader so drill home the target-of-attention more deeply, not leaving any room in the reader’s mind that this story is about FTO and IPR.

 Methods

FTO appears for the first time in text as an abbreviation. Scientific English convention requires abbreviations and acronyms to be spelled out in full words, then followed by abbreviation in parentheses (FTO).

Line 59 – could authors explain how the scientific quality and validity were assessed?

Can authors add details on whether any papers were excluded from review, i.e., the exclusion criteria, the total number of papers assessed, the tools to extract the papers and or any applicable data?

 Results

Line 78 – could authors provide a brief explanation of DNA CpG island and implications to gene expression following its methylation?

Lines 100-106 – Result interpretation could be improved. What do the % methylation levels represent? Did bovine milk under-methylate when compared to human milk?

Line 108 – FTO definition should be moved above to appear earlier in text, at first occurrence.

Line 121 – SNPs not defined. Do SNPs affect the methylation of FTO? How are SNPs and methylation connected in this section?

Line 154 – errant bracket. 

Line 401 – errant space.

Figure 2 provides an excellent overview of the core topics in review. Thank you for this thoroughness.

Lines 787-788 – for published work, have authors checked Yan, X.; Liu, L.; Yao, S.; Chen, Y.; Yu, Q.; Jiang, C.; Chen, W.; Chen, X.; Han, S. LncRNA and MRNA Profiles of Human Milk-Derived Exosomes and Their Possible Roles in Protecting against Necrotizing Enterocolitis. Food Funct. 2022, 13 (24), 12953–12965. https://doi.org/10.1039/D2FO01866G.

 It may be relevant to include human tissue and cell specific expression of FTO and postulate where it may be affected in infants consuming breast milk or formula. See, for example https://www.proteinatlas.org/.

 Conclusion

Lines 872-876 – The research that may have provided some basis to the statements made here rely on correlative studies. The words “early cause” may therefore also be too strong to use. The final statement also does not inform the general community, since breastfeeding is not always available in neonatal intensive care units, and not all mothers are able to breastfeed. Rather, the approach towards improvement of formula from the perspective of additives to target FTO-related pathways may provide a more relevant concluding statement, especially from a public health perspective.

 Overall

Suggestion to exclude dramatic terms, and to limit the use of more colloquial words, including “apparently, obviously”.

 Authors should briefly discuss the reversibility of DNA methylation and modifications in the context of a link between postnatal feeding to adulthood obesity. Would diet-induced modification during the postnatal period persist to adulthood? Or are they reinforced, further modified, during one’s lifetime diet?

 Since the development of diseases is often multi-factorial, it is conflicting to not only pin down the disease to epigenetics, but then further narrow it to formula feeding versus breast milk only. Authors should mention whether the studies looking at postnatal feeding patterns and disease outcomes considered the child’s diet once solid foods were introduced. Whether a child is fed processed foods may also influence disease development, potentially epigenetically.

 Additionally, microbiome is considered a major factor in infant and child health outcomes, which is heavily influenced by nutrition. Could changes in the microbiome affect FTO?

Comments on the Quality of English Language

Adequate, but ther is no room for 'obviously' or actually or evidently..... More effort could be employed to present a crisper clearer English with less verbosity, more direct statements of fact. No drama in scientific English.

Author Response

Response to Reviewer 2

We appreciate all the valuable comments and highly constructive contributions from R2, which greatly helped to improve the quality of our review article.

Abstract

R2 requested that we should clarify the purpose of our review.

Thank you for pointing that out. The updated abstract now addresses this previously overlooked aspect.

In order to provide a more succinct overview of our literature search, we have included a statement indicating that we searched through the Web of Science, Google Scholar and PubMed databases, as recommended by R2.

Introduction

R2 suggested providing more background information on FTO.

There are numerous articles discussing the role of FTO in cell function and various diseases. A search for “FTO” on PubMed yields 4885 papers, while searching for “fat mass- and obesity-associated protein”, results in an even higher number of 6074 publications.

In order to focus on the role of FTO in adipogenesis and obesity in relation to postnatal feeding patterns (breastmilk versus formula), we provided extensive information on this issue, citing a total of 339 articles in our review.

Nevertheless, in the revised version we emphasized that

m6A modifications have attracted recent attention due to their pivotal importance in regulating adipogenic gene expression [60-62].

  1. Ben-Haim, M.S.; Moshitch-Moshkovitz, S.; Rechavi, G. FTO: linking m6A demethylation to adipogenesis. Cell Res. 2015, 25, 3-4.
  2. Zhao, X.; Yang, Y.; Sun, B.F.; Shi, Y.; Yang, X.; Xiao, W.; Hao, Y.J.; Ping, X.L.; Chen, Y.S.; Wang, W.J.; et al. FTO-dependent demethylation of N6-methyladenosine regulates mRNA splicing and is required for adipogenesis. Cell Res. 2014, 24, 1403-1419.
  3. Azzam, S.K.; Alsafar, H.; Sajini, A.A. FTO m6A demethylase in obesity and cancer: Implications and underlying molecular mechanisms. Int. J. Mol. Sci. 2022, 23, 3800.

Recent studies have shown that m6A marks are decreased in the adipose tissue of obese subjects. These marks are involved in regulating obesity-associated processes, such as adipogenesis, lipid metabolism, and insulin resistance [50,63,64].

  1. Yang, Z.; Yu, G.L.; Zhu, X., Peng, T.H.; Lv, Y.C. Critical roles of FTO-mediated mRNA m6A demethylation in regulating adipogenesis and lipid metabolism: Implications in lipid metabolic disorders. Genes Dis. 2021, 9, 51-61.
  2. Sun, M.; Zhang, X. Epigenetic regulation of N6-methyladenosine modifications in obesity. J. Diabetes Investig. 2021, 12, 1306-1315.
  3. Wu, R.; Wang, X. Epigenetic regulation of adipose tissue expansion and adipogenesis by N6-methyladenosine. Obes. Rev. 2021, 22, e13124.

Excellent recent publications provide extensive information on FTO and its impact on m6A-dependent RNA expression [78-84]

  1. Yue, Y.; Liu, J.; He, C. RNA N6-methyladenosine methylation in post-transcriptional gene expression regulation. Genes Dev. 2015, 29, 1343-1355.
  2. Erson-Bensan, A.E.; Begik, O. m6A Modification and Implications for microRNAs. Microrna 2017, 6, 97-101.
  3. Bartosovic, M.; Molares, H.C.; Gregorova, P.; Hrossova, D.; Kudla, G.; Vanacova, S. N6-methyladenosine demethylase FTO targets pre-mRNAs and regulates alternative splicing and 3'-end processing. Nucleic Acids Res. 2017, 45, 11356-11370.
  4. Huang, J.; Yin, P. Structural insights into N6-methyladenosine (m6A) modification in the transcriptome. Genomics Proteomics Bioinformatics 2018, 16, 85-98.
  5. Zhu, Z.M.; Huo, F.C.; Pei, D.S. Function and evolution of RNA N6-methyladenosine modification. Int. J. Biol. Sci. 2020, 16, 1929-1940.
  6. He, P.C.; He, C. m6 A RNA methylation: from mechanisms to therapeutic potential. EMBO J. 2021, 40, e105977.
  7. Khan, .FA.; Nsengimana, B.; Awan, U.A.; Ji, X.Y.; Ji, S., Dong, J. Regulatory roles of N6-methyladenosine (m6A) methylation in RNA processing and non-communicable diseases. Cancer Gene Ther. 2024, Epub ahead of print. doi: 10.1038/s41417-024-00789-1

Other excellent and comprehensive reviews that highlight the impact of FTO in adipogenesis and lipid metabolism are cited in the text

  1. Yeo, G.S.; O'Rahilly, S. Uncovering the biology of FTO. Mol. Metab. 2012, 1, 32-36.47.
  2. Yang, Z.; Yu, G.L.; Zhu, X., Peng, T.H.; Lv, Y.C. Critical roles of FTO-mediated mRNA m6A demethylation in regulating adipogenesis and lipid metabolism: Implications in lipid metabolic disorders. Genes Dis. 2021, 9, 51-61.
  3. Zhao, X.; Yang, Y.; Sun, B.F.; Zhao, Y.L.; Yang, Y.G. FTO and obesity: mechanisms of association. Curr. Diab. Rep. 2014, 14, 486.
  4. Deng, X.; Su, R.; Stanford, S.; Chen, J. Critical enzymatic functions of FTO in obesity and cancer. Front. Endocrinol. (Lausanne) 2018, 9, 396.

We believe that the interested reader can easily find more background information on FTO´s involvement in lipid metabolism and adipogenesis in the cited papers.

Methods

R2 required that abbreviations and acronyms should be spelled out in full words, followed by the abbreviation in parentheses (FTO).

We unified the text accordingly.

We provided additional information on paper selection and the exclusion of redundant publications. The quality of the selected papers was evaluated by all authors, who are experts in molecular biology, particularly in the fields of lipid metabolism, molecular genetics, and epidemiological research.

Results

R2 asked for a brief explanation of DNA CpG islands and their implications for gene expression following methylation.

In line 84/85 we included the statement that “CpG islands are DNA methylations regions in promoters that silence corresponding gene expression”

R2 was curious about the percentage of methylation levels in genes and gene promoters that are represented after feeding rats milk from different animals. It was found that the consumption of bovine milk leads to lower levels of methylation compared to human milk. We added the following sentence to the study: “These data point to differences in the capacity and magnitude of promoter and whole gene body methylations between cow milk and human milk intake” For more details see ref. 39

Trinchese, G.; Feola, A.; Cavaliere, G.; Cimmino, F.; Catapano, A.; Penna, E.; Scala, G.; Greco, L.; Bernardo Porcellini, A.; Crispino, M.; et al. Mitochondrial metabolism and neuroinflammation in the cerebral cortex and cortical synapses of rats: effect of milk intake through DNA methylation. J. Nutr. Biochem. 2024, 128, 109624.

R2 found that the FTO definition should be moved up to appear earlier in the text, at its first occurrence.

Thank you, we have made the correction.

R2 claimed that SNPs were not defined. Furthermore, R2 asked whether SNPs affect the methylation of FTO. How are SNPs and methylation connected in this section?

In the revised version, we provided the definition of single nucleotide polymorphisms (SNPs). We found evidence that SNPs do indeed modify FTO methylation, which was a very interesting point raised by R2. We have included the new reference 45:

  1. Bell, C.G.; Finer, S.; Lindgren, C.M.; Wilson, G.A.; Rakyan, V.K.; Teschendorff, A.E.; Akan, P.; Stupka, E.; Down, T.A.; Prokopenko, I.; et al. Integrated genetic and epigenetic analysis identifies haplotype-specific methylation in the FTO type 2 diabetes and obesity susceptibility locus. PLoS One 2010, 5, e14040.

R2 suggested that it may be relevant to include the specific expression of FTO in human tissue and cells when comparing breast feeding versus formula feeding.

This is of high scientific importance and could easily help resolve the black box of epigenetics induced by formula feeding. However, it is not ethically possible to take tissue samples from healthy newborn babies to investigate these differences. The maximum that can be done is to study blood mononuclear cells, as published by Cheshmeh et al. Primate studies may serve as an approximation for future research.

We should not forget that formula was introduced WITHOUT ANY DATA on epigenetics a century ago. From an objective scientific perspective, formula feeding is an open field experiment on human infants, which for instance would never be allowed when a new drug is introduced for medical treatment.

Conclusion

R2 finds our emphasis on deviated postnatal FTO signaling leading to adipogenic stem cell differentiation as an “early cause” of obesity to be TOO strong. We believe that this is the KEY CONCLUSION of the manuscript. FTO is involved in regulating a critical postnatal window of epigenetic imprinting. Gain-of-function SNPs highlight the impact of increased protein-stimulated FTO signaling on BMI trajectories in adolescence. The paper includes many more observations that support our perspective.

Our intention is not to focus on improving formula, but rather to revisit human physiology during the postnatal period of metabolic programming. Formula should only be used as a last resort for mothers who are unable to breastfeed. By making formula the norm, we are contributing to our current metabolic epidemics. Cesarean sections, initially meant for emergency situations, have now become the norm in over 50% of deliveries in developed countries.

For mothers who are unable to breastfeed, the first alternative should be human donor milk derived from a milk bank providing the closest replica of human milk.

Formula can be improved by the addition of milk exosomes, especially for infants with necrotizing enterocolitis. For these infants, colostrum is the best option. The milk of mothers who give birth prematurely contains much higher levels of miR-22-3p and miR-148a-3p as outlined in our review. How will formula be able to replicate the dynamics of milk exosomes and their cargo?

From a public health perspective, we strongly support the Lancet Breastfeeding Group in their efforts to promote a return to physiological breastfeeding whenever possible.

Overall

R2 requested that we to exclude dramatic terms and colloquial words.

We revised the manuscript accordingly.

R2 also wondered whether modifications induced by diet during the postnatal period would persist into adulthood or if they would be further reinforced or modified throughout one’s lifetime due to diet.

This is an important question. We have presented evidence that increased FTO-mTORC1-S6K1-signaling, which epigenetically suppresses WNT signaling, may be a potential driving mechanism of aberrant early adipogenic stem cell differentiation. To support this concept, we have included additional literature data and introduced a new Figure (now Figure 2 in the revised version). During the postnatal period, there needs to be a balance between BAT and WAT development. BAT is crucial for infants´ thermoregulation and is associated with lower FTO signaling, whereas WAT development is driven by increased FTO signaling.

Of course, after the postnatal period, diet will have an impact on tissue methylation at the level of DNA and RNA. However, the first 1000 days from conception are regarded as critical for metabolic programming linking disturbances during this period to the onset of later non-communicable diseases, especially obesity and diabetes. Infants fed protein-rich formula have a higher predisposition to obesity in childhood and adolescence, highlighting the significant impact and vulnerability of the postnatal period for metabolic programming.

In the revised version we addressed this issue accordingly.

R2 addressed the potential influence of the microbiome on FTO signaling. This is a very interesting aspect and indeed, we found some indications for the microbiome-FTO interaction. Gut bacteria and their metabolites might interfere with FTO signaling. An interesting field for future research!

Additionally, milk exosomes have been shown to interfere with the growth and selection of gut bacteria, potentially preventing necrotizing enterocolitis. This information was appropriately cited at the end of the paper.

In our own pilot study, we discovered that orally administered bovine milk exosomes to a genetic mouse model of ulcerative colitis significantly reduced intestinal inflammation (Stremmel W, Weiskirchen R, Melnik BC. Milk Exosomes Prevent Intestinal Inflammation in a Genetic Mouse Model of Ulcerative Colitis: A Pilot Experiment. Inflamm Intest Dis. 2020;5(3):117-123. doi: 10.1159/000507626).

Reviewer 3 Report

Comments and Suggestions for Authors

The review is appropriately written; however, there are some issues before it can be published.

Abstract: OK

Introduction: short it should be expanded. A few phrases about  m6A RNA and its implications.

Material and methods: OK

Results: At 3.2, please add a concise schematic (Drawing) of the issues discussed in the paragraph.

The list of genes upregulated by FTO-mediated m6A mRNA demethylation   should be converted to a concise schematic drawing.

              3.10 given the importance of zing finger proteins – a summarizing drawing scheme- at the end must be included.

 Discussion: a short paragraph regarding the latest developments in the field regarding the discussed issue should be added. The paragraph can be in the form of a summarizing table ( with proper citation) .

Also, a phrase about the strong points and the weak points of the review must be added.

 Conclusion: OK 

Author Response

Response to Reviewer 3

We would like to express our gratitude for R3 for all comments and suggestions.

Introduction

R3 requested additional information on m6A RNA and its implications.

We have included these phrases in both the introduction and results sections. See comments for R2.

Results

R3 has requested a brief schematic drawing outlining the issues discussed in section 3.2. Additionally, R3 recommends converting the list of genes upregulated by FTO-mediated m6A mRNA demethylation, as well as zinc finger protein (ZNF), into a concise schematic drawing.

To enhance the clarity of our general message, a new Figure 2 has been added. This figure illustrates the ZNF217-FTO-mTORC1-S6K1-WNT signaling pathway, as well as the FTO-m6A-mediated demethylation of MYC and CEBPD. These processes are crucial in the early differentiation of adipocyte stem cells.

Discussion

R3 has requested a brief paragraph outlining the most recent advancements in the field related to the discussed issue to be included.

As far as we know, we are leading the way in FTO signaling research in infant feeding. The most recent paper on this topic was published by Cheshmeh et al. in 2020 (see ref. 65).

We have expanded our conclusion to highlight the necessary future developments in the field concerning the impact of the microbiome on FTO signaling as suggested by R2.

Additionally, R3 asked for a discussion on the strengths and weaknesses of the review, which we have included. We have also acknowledged the limitations of our review in the Discussion section.

Round 2

Reviewer 3 Report

Comments and Suggestions for Authors

The manuscript has been considerably improved improved.